# Extending Fourier neural operators for parameterized and coupled PDEs

**Cheng Jing**[1]* **Uvini Balasuriya**[1]* **Abhishek Verma**[2] **Kallol Bera**[2] **Shahid Rauf**[2] **Kookjin Lee**[1]
[1]Arizona State University,    [2]Applied Materials Inc.
{cjing6, ubalasur, Kookjin.Lee}@asu.edu,
{AbhishekKumar_Verma, Kallol_Bera, Shahid_Rauf}@amat.com

## Abstract

Parameterized and coupled partial differential equations (PDEs) are central to modeling phenomena in science and engineering, yet neural operator methods that address both aspects remain limited. We extend Fourier neural operators (FNOs) with minimal architectural modifications along two directions. For parameterized dynamics, we propose a hypernetwork-based modulation that conditions the operator on physical parameters. For coupled systems, we conduct a systematic exploration of architectural choices, examining how operator components can be adapted to balance shared structure with cross-variable interactions while retaining the efficiency of standard FNOs. Evaluations on benchmark PDEs, including the one-dimensional capacitively coupled plasma equations and the Gray–Scott system, show that our methods achieve up to 55∼72% lower errors than strong baselines, demonstrating the effectiveness of principled modulation and systematic design exploration.

## 1 Introduction

Numerical simulation has long served as a cornerstone of scientific and engineering inquiry, underpinning advances in areas ranging from fluid dynamics (Ferziger et al., 2019) and climate modeling (Bauer et al., 2015) to material science (Rappaz et al., 2003) and structural analysis (Macneal & Harder, 1985). High-fidelity simulations, while indispensable, often entail substantial computational cost. These costs become especially significant when conducting parameter studies, uncertainty quantification, and real time decision making. As a result, surrogate modeling has emerged as an essential tool to expedite simulation workflows, enabling rapid approximations that preserve first-order fidelity while dramatically reducing computational burden. Recently, deep learning techniques have gained increasing traction in this domain. Methodologies such as physics-informed neural networks (PINNs) (Raissi et al., 2019; Karniadakis et al., 2021) and neural operators (NOs) (Li et al., 2020a;b; Lu et al., 2021) offer flexible, data-driven approximations of parametrized partial differential equations (PDEs), often delivering orders-of-magnitude speedups over traditional solvers while maintaining acceptable accuracy.

Despite significant progress in NO, most existing studies have considered scenarios with both varying initial conditions and parameter regimes, yet they often overlook the particular, yet prevalent case of parameterized dynamics with fixed initial conditions, typified by equations of the form $u_t = f(u; \mu)$ with $u(0) = u_0$. Such settings are ubiquitous in engineering, where model dynamics depend on physical parameters (e.g., forcing terms in inviscid Burgers' equation (Rewieński, 2003; Carlberg et al., 2013), diffusivity/reaction constants in chemically reacting flows (Buffoni & Willcox, 2010; Lee & Carlberg, 2020), and Reynolds number and viscosity in fluid dynamics (Stabile & Rozza, 2018)) while initial states remain unchanged across simulations. Studying this regime is important for practical applications, for example, to understand how variations in material properties, operating conditions, or control parameters affect system behavior from a fixed, well-defined state.

In this work, we focus specifically on NOs for parameterized coupled PDEs in fixed-initial-condition settings. To facilitate this exploration, we build upon Fourier neural operators (FNOs) (Li et al.,

---

*Equal Contribution

2021), a method now well-established in the community for its expressive parametric kernel in Fourier space. Our approach is designed to remain general and modular, making it adaptable to both FNO variants and other neural operator frameworks. To incorporate parameterized dynamics effectively, we introduce an architectural modification in which the meta-physics-knowledge encoded in the governing equations is captured by a larger set of shared base parameters, while parameter-specific variations are represented through a smaller set of task-dependent model parameters (with a task being associated a specific physical parameter). To ensure the overall model remains lightweight, we implement this design using a compact hypernetwork, enabling parameterization while retaining a model size comparable to standard, non-parameterized FNOs.

Further, we extend the (parameterized) FNOs to effectively model the coupled systems while being efficient in the model size. While existing efforts (Xiao et al., 2023) introduce specialized decompositions stream to capture cross-field interactions, there is little systematic guidance of the design principles that govern effective cross-variable coupling in neural operators, particularly when aiming to preserve the efficiency and scalability of standard FNOs. In this paper we address that gap: we present a principled design space for extending FNOs to coupled problems, investigate the lift, Fourier layers, and projection operators for identifying effective locations and mechanisms for enabling cross-variable coupling.

In addition, we introduce a new benchmark PDE based on a simplified one-dimensional capacitively coupled plasma (CCP) model. This system is of practical importance in plasma physics and semiconductor manufacturing, and it exhibits rich parameterized dynamics that challenge existing operator-learning methods. Our formulation provides a tractable yet representative setting for evaluating neural operators under coupled and parameterized conditions.

Our contributions are summarized as

- Parametric extensions of FNOs for effectively modeling parameterized dynamics,
- Systematic dissection and adaptation of FNO architecture components for coupled systems,
- Introduction of a novel benchmark problem with many physical parameters describing simplified sheath dynamics governed by plasma physics, and
- Extensive experimentation with the proposed methods, comparisons with baselines, and validation on benchmark problems.

## 2  TECHNICAL BACKGROUND

### 2.1  NEURAL OPERATORS

NOs are a type of data-driven surrogate model designed to learn mappings between functions rather than traditional input-output pairs, making them particularly effective for problems governed by PDEs (Lu et al., 2021; Kovachki et al., 2023; Boullé & Townsend, 2024; Azizzadenesheli et al., 2024). By taking discrete representations of continuous functions as input, NOs produce functional representations that can efficiently approximate and simulate complex physical systems.

Let $\mathcal{A}$ and $\mathcal{U}$ be two function spaces, and consider a nonlinear operator $G : \mathcal{A} \to \mathcal{U}$ mapping between them. Neural operators are a class of machine learning methods designed to construct a surrogate $G_\Theta \approx G$ within a trial space of neural networks. Given input-output pairs $\{(a^i, u^i)\}_{i=1}^n$, where $x^i \in \mathcal{A}$ and $u^i = G(a^i) \in \mathcal{U}$, a neural operator can be trained in the standard supervised manner by solving: $\min_\Theta \sum_{i=1}^n L\left(G_\Theta(a^i), u^i\right)$, where $L$ denotes a loss term and $\Theta$ denotes the parameters of the neural operator $G_\Theta$.

In practice, the functional input data $a^i$ are first discretized; for example, if $a(x)$ and $u(x)$ are functions, the grid representation $\boldsymbol{a} = [a(x_1), \ldots, a(x_m)]^\mathsf{T} \in \mathbb{R}^{m \times d_a}$ serves as input along with an evaluation point (or set of points) $D = \{x_1, \ldots, x_m\}$ in the domain of $a$, and the neural operator learns a mapping $\boldsymbol{a} \mapsto G_\Theta(\boldsymbol{a})$ that predicts $G_\Theta(\boldsymbol{a})(x_p) \approx u(x_p)$ (Lu et al., 2021; Li et al., 2021). With the assumption that the spatial grid where $a(x)$ and $u(x)$ are discretized is fixed, $(D, \boldsymbol{a}) \mapsto \boldsymbol{u}$ (under $G$) produces a vector $\boldsymbol{u}$ of operator evaluations, and therefore samples $(\boldsymbol{x}, \boldsymbol{a}, \boldsymbol{u})$ form the training data for the NO learning problem.

## 2.2 Fourier neural operators

FNOs are a specific instantiation of NOs that model learnable nonlinear kernel integral operators, with the integrals efficiently evaluated in the Fourier frequency domain. FNOs, $G_\theta : \mathcal{A} \to \mathcal{U}$, first projects the input data into hidden representations via local transformation, $v_0(x) = P(a(x))$, where $P$ is typically chosen as a linear projector in practice. FNOs then employ the iterative updates of hidden representations $v_\ell \mapsto v_{\ell+1}$ via

$$v_{\ell+1}(x) \coloneqq \sigma\Big( W v_\ell(x) + \big( \mathcal{K}(a; \phi) v_\ell \big)(x) \Big), \qquad \forall x \in D \tag{1}$$

where $W$ is a linear transformation and $\mathcal{K}$ denotes a kernel integral operator. The kernel integral operator is defined as

$$(\mathcal{K}(a; \phi) v_\ell)(x) \coloneqq \int_D \kappa(x, y, a(x), a(y); \phi) v_\ell(y) \mathrm{d}y, \qquad \forall x \in D \tag{2}$$

where $\phi$ indicates the learnable parameters that characterize the kernel. For efficient computation of Eq. equation 2, FNOs define the kernel integral operator in Fourier space and perform a convolution operation in that space such that

$$(\mathcal{K}(\phi) v_\ell)(x) = \mathcal{F}^{-1}\Big( R_\phi \cdot (\mathcal{F} v_\ell) \Big)(x), \qquad \forall x \in D, \tag{3}$$

where $\mathcal{F}$ and $\mathcal{F}^{-1}$ denote Fourier and inverse Fourier transformations, respectively, and $R_\phi$ denotes the Fourier transform of function $\kappa$. In practice, $R_\phi$ is parameterized as a learnable tensor. The last hidden representation $v_T(x)$ is then projected back to the target data space via a local transformation, $u(x) = Q(v_T(x))$, where $Q$ is typically modeled as a shallow multi-layer perceptron (MLP).

## 3 Related work

Building on the success of FNOs, there have been many variants extending their capabilities. For example, physics-informed neural operator (PINO) (Li et al., 2024) and Geo-FNO (Li et al., 2023) extend FNO with physical consistency or geometric generality.

More closely related to our work are efforts that enhance FNO architectures to improve expressivity and efficiency. Factorized FNO (Tran et al., 2023), U-FNO (Wen et al., 2022), and U-NO (Rahman et al., 2023) propose structural modifications: separable spectral layers or encoder-decoder designs that enable stable training. Our approach follows a similar philosophy of architectural enhancement, but with emphasis on parameterized and coupled systems rather than general-purpose scalability.

Within the FNO family, explicit modeling of physical parameters remains relatively underexplored. The recently proposed HyperFNO (Alesiani et al., 2022) infers the model parameters of FNO via a hypernetwork conditioned on known physical parameters, illustrating the benefit of encoding parametric dependencies. Another complementary study (Subramanian et al., 2023) explores scaling laws and transfer behavior of neural operators under large datasets and compute budgets. Our method shares the goal of parameter-aware modeling but focuses on a compact modulation mechanism that preserves the lightweight character of standard FNOs.

For coupled systems, prior works such as coupled multiwavelet neural operator (CMWNO) (Xiao et al., 2023) and attention-based multiphysics models (Rahman et al., 2024) have introduced cross-field coupling through wavelet or large-scale pre-training. In contrast, our work provides a systematic and minimal architectural extension to FNOs, achieving efficient cross-variable interaction without sacrificing model compactness.

## 4 Methods

Our methods seek minimal yet principled architectural modifications to FNOs that enable them to handle parameterized and coupled dynamical systems. We emphasize minimal modification as a design principle: this ensures (i) the resulting models preserve the efficiency and scalability of FNOs and (ii) performance gains can be attributed to structural insights rather than increased capacity.

To this end, we focus on the three core computational components of FNOs: the lift operator (input projection), the Fourier layers (spectral updates), and the projection operator (output mapping). We systematically explore how to adapt each component to encode parameter dependence and to capture cross-variable interactions in coupled systems. Section 4.1 presents the parameterized extension via hypernetwork-based modulation, and Section 4.2 details the coupled extension by introducing the design spaces and spectral-domain coupling.

## 4.1 Extensions to Parametrized FNO

A compact hypernetwork-based modulation scheme is proposed to extend FNOs for handling physical parameters, $\mu \in \mathbb{R}^{n_\mu}$. A lightweight hypernetwork (Ha et al., 2017) takes the physical parameters $\mu$ as input and outputs a set of shifts, $s(x, \mu) = f^{\text{hyper}}(x, \mu)$, where $s(x, \mu) = [s_1(x, \mu), \ldots, s_L(x, \mu)]$ corresponds to layer-specific adjustments. These shifts are incorporated as additional biases in the Fourier layers:

$$v_{\ell+1}(x) := \sigma\Big(W v_\ell(x) + \left(\mathcal{K}(a; \phi) v_\ell\right)(x) + s_\ell(x, \mu)\Big). \qquad \forall x \in D \qquad (4)$$

We refer to this variant using "hp" (e.g., `hpFNO`) indicating the hypernetwork-based approach (Figure 1).

The rationale behind this design is twofold. First, modulation has already proven effective in implicit neural representations (INRs) (Sitzmann et al., 2020; Fathony et al., 2020; Dupont et al., 2022) and physics-informed neural networks (PINNs) (Cho et al., 2023), where conditioning on auxiliary variables enables networks to represent families of functions without retraining. Second, unlike a simple input augmentation, modulation operates directly on internal representations, allowing parameter dependence to influence the dynamics at every layer rather than only at the input. Together, these properties make `hp`-variant a lightweight yet expressive model for parameterized dynamics.

Compared to HyperFNO (Alesiani et al., 2022), which infers a large part of the base FNO through a hypernetwork, our approach adopts a lightweight shift modulation that perturbs intermediate activations while keeping the core weights fixed. Within the considered scenarios in this work, this design achieves comparable flexibility with substantially lower parameter overhead and complexity.

For completeness, we also consider simpler variant that employs input augmentation to encode physical parameters. In this approach, the parameters are concatenated with the function input as $[a(x); \mu(x)]$, where $\mu(x) = \mu$ is constant across the spatial domain. The lift operator then encodes this augmented input into a latent representation, $v_0(x) = P([a(x), \mu(x)])$. This strategy mirrors prior work, for example in the neural ODE community, where augmenting the input has been shown to enhance the expressivity of the learned dynamics (Dupont et al., 2019; Massaroli et al., 2020; Lee & Parish, 2021), and PINNs (Cai et al., 2021; Cho et al., 2024). We refer to this variant using the prefix "p" (e.g., `pFNO`) to denote parameterization via input augmentation.

While input augmentation is straightforward and effective in some cases, it may not fully capture how physical parameters influence the underlying dynamics. The hypernetwork-based modulation therefore provides a more expressive yet efficient alternative, conditioning the operator throughout its layers rather than only at input. Within our experiments, the hypernetwork-based approach consistently achieves better performance without significantly increasing model complexity.

## 4.2 Extensions to coupled FNO

We now turn to extending FNOs for coupled systems, where multiple interdependent variables evolve under shared dynamics. For simplicity of exposition, we focus on the case of two variables and denote each component by superscript, $z^\square$ for a generic variable $z$, where $\square \in \{\alpha, \beta\}$. With this notation, the operator of interest becomes $(u^\alpha(x), u^\beta(x) = G_\Theta(a^\alpha(x), a^\beta(x))$. Our design philosophy follows the same principle outlined at the beginning of this section: to retain the minimal and scalable structure of standard FNOs while introducing only the modifications necessary to capture cross-variable interactions. To this end, we introduce coupling in the spectral domain, where the spectral representations of multiple variables are mixed through a lightweight encoder-decoder mechanism. To support this coupling, we explore how the core FNO computational components, namely the lift operator, $P$, Fourier layers, and projection operator, $Q$, can be configured, including choices on whether those operators are shared or defined separately across variables.

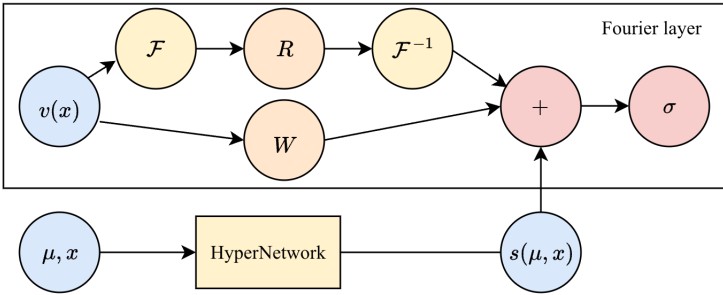

Figure 1: The proposed `hpFNO` architecture. The hypernetwork produces a parameter-dependent 'shift' term, $s(\mu, x)$, which acts as a parameter-dependent bias term in each Fourier layer.

**Lift operator, $P$** For the local transform, $(Pa)(x) = v_0(x) \in \mathbb{R}^{d_v}$, we consider two main design choices that differ in whether the operator is shared or individual across the two variables.

(P1) (Shared): A single operator $P : \mathbb{R}^1 \to \mathbb{R}^{d_v}$ is shared across the two input components, so that both variables are mapped into the latent space using the same transformation.
(P2) (Separate): Two individual mappings, $P = (P^\alpha, P^\beta)$, allow variable-specific latent representation.

**Fourier layers (spectral coupling)** The main modification enabling cross-variable interaction occurs in the Fourier layers, which consist of two components: the point-wise linear map $Wv_\ell(x)$ (L1 / L2) and the global spectral convolution $\mathcal{K}v_\ell(x)$ (G).

(L1) / (L2): The point-wise linear map can be implemented either as a single operator shared by both variables (L1) or as two separate operators ($W^\alpha$ and $W^\beta$) (L2).

(G) For coupled systems, we define the global spectral convolution to perform coupling only in Fourier space. Specifically, each variable is first transformed independently, $\tilde{v}^\square(k) = \mathcal{F}v_\ell^\square(k)$, for $\square = \{\alpha, \beta\}$. Their spectral representations are then combined through a shallow encoder network $f^{\text{enc}}(\tilde{v}^\alpha(k), \tilde{v}^\beta(k))$, followed by the usual mode-wise kernel multiplication $\tilde{\tilde{v}}(k) = R_\phi(k)\tilde{v}(k)$. The result is decomposed into variable-specific coefficients using another shallow network, $(\tilde{\tilde{v}}^\alpha(k), \tilde{\tilde{v}}^\beta(k) = f^{\text{dec}}(\tilde{\tilde{v}}(k))$, and finally mapped back to the data space via the inverse Fourier transform, $v_{\ell+1}^\square(k) = \mathcal{F}^{-1}(\tilde{\tilde{v}}^\square(k))$, for $\square = \alpha, \beta$.

A key design choice here is that coupling is introduced only in Fourier space, after the individual Fourier transforms. This has several benefits compared to approaches that use entirely separate convolution operators and exchange hidden representations across variables. First, it preserves the core structure of FNOs, keeping the model lightweight and scalable. Second, the Fourier domain naturally captures long-range correlations, making it a well-suited location for cross-variable mixing. Finally, by avoiding repeated exchanges of hidden states in the spatial domain, this design reduces computational overhead while maintaining symmetry between the two variables.

**Projection operator, $Q$** The local transform, $(Qv_T)(x) = u(x)$, resembles with that of the lift operator $P$ and, thus, similar choices apply. We focus here on the single-channel output format for each variable, which is the natural setting for coupled systems.

(Q1) (Shared): A single projection is shared by both outputs, $Q : \mathbb{R}^{d_v} \to \mathbb{R}^1$.
(Q2) (Separate): Two operators are separately defined for each output component, $Q = (Q^\alpha, Q^\beta)$, with $Q^\square : \mathbb{R}^{d_v} \to \mathbb{R}^1$ for $\square \in \alpha, \beta$.

To refine the design space under (Q2), we adopt the adaptive basis viewpoint (Cyr et al., 2020), which interprets the last hidden activation as a set of adaptive basis functions, $\Psi(a(x)) = [\psi_1(a(x)), \ldots, \psi_{n_q}(a(x))]^\top$, and the weight of the output layer as coefficients, $\Xi = [\xi_1, \ldots, \xi_{n_q}]^\top$: $u(x) = \sum_{i=1}^{n_q} \xi_i \psi_i(x)$. In practice, $Q$ is parameterzed as a shallow MLP, $(Qv_T)(x) = W_2\sigma(W_1v_T(x) + b_1) + b_2$ and the adaptive basis corresponds to $\Psi(a(x)) = \sigma(W_1v_T(x) + b_1)$

and the coefficients to $\Xi = W_2$. The shared/separate choices are then realized by sharing or splitting the weight matrices and biases.

From this perspective, $Q$ can differ in whether the basis and the coefficients are shared or separate: (Q2a) Shared basis and separate coefficients, (Q2b) Separate basis and shared coefficients, and (Q2c) Separate basis and separate coefficients.

### 4.3 PUTTING ALL TOGETHER

Based on our experimentation (presented in Appendix D.1.1) with the different combinations of the design choices (further elaborated in Appendix A), we define our model, extended FNOs — $\mathrm{FNO_x}$, as a realization of the combination: (P1) + (L2) + (G)+ (Q2c), which employs the shared lift operator, the separate point-wise linear maps, the proposed coupled global spectral convolution layer, and the separate sets of basis functions and separate sets of coefficients in the projection operator.

## 5 EXPERIMENTAL RESULTS

In this section, we present experimental results comparing the proposed methods with a range of baseline models. All methods are implemented in PYTHON with PYTORCH (Paszke et al., 2019), building upon the original implementation of FNOs (Li et al., 2021) and extending it to handle coupled and parameterized settings. Each experiment is repeated five times with different random seeds, and all computations are performed on an NVIDIA A100 GPU with 80GB memory.

### 5.1 SETUP

**Tasks and training** We consider time-dependent PDEs that generate solution trajectories $\{u(x, t; \mu)\}_{t=0}^T$ with $u(x, 0; \mu) = u(0)$. NOs are formulated as one-step evolution maps that approximate the discrete-time flow of the underlying PDE. Specifically, given a finite history of past solution states $\{u(t-\tau)\}_{\tau=0}^{T_{\mathrm{in}}-1} \in \mathcal{A}$, the operator predicts the next state $u(t+1) \in \mathcal{U}$: $G_\Theta : \mathcal{A} \to \mathcal{U}$. Multi-step forecasts are obtained autoregressively by recursively applying the same operator. During training and testing, the initial input window $(u(T_{\mathrm{in}} - 1), \ldots, u(0))$ is assumed to be available, while subsequent rollouts incorporate predicted states back into the input window.

**Baselines** As baselines of comparisons, we consider methods inherited from Xiao et al. (2023):

- $\mathrm{FNO_c}$: FNOs taking inputs as a vertical concatenation of the discretized input data,
- $\mathrm{CFNO}$: Two separate FNOs sharing exchanging hidden representation in Fourier layers,
- $\mathrm{MWT_c}$: MWTs (Gupta et al., 2021) taking a vertically concatenated input data,
- $\mathrm{CMWNO}$: Multiwavelet NOs specifically designed for coupled systems (Xiao et al., 2023),

a FNO baseline considering parametric extension via hypernetworks (Alesiani et al., 2022):

- $\mathrm{HyperFNO_c}$: FNOs whose model parameters of the lift/projection operators (P/Q), the pointwise linear map, $W$, and the kernel weight $R$ are inferred from hypernetworks,

other relevant baselines, DeepONets (Lu et al., 2021) and U-Nets (Ronneberger et al., 2015), modified to handle multi input:

- $\mathrm{DON_c}$: DeepONets extended to take a vertical concatenation of the discretized input data,
- $\mathrm{U\text{-}Net_c}$: U-Nets taking two-channel input,

and, finally, one variant of a certain combinations of choices from the described design choices:

- $\mathrm{FNO_m}$: the lift operator projects the two-channel input $(a^\alpha, a^\beta)$ jointly into a shared latent space, i.e., $P : \mathbb{R}^2 \to \mathbb{R}^{d_v}$ with $v_0(x) = P([a^\alpha(x), a^\beta(x)])$.

We refer to Appendix B.1 for the detailed description of the baselines.

**Performance evaluation** We evaluate performance using the normalized root mean square error (nRMSE), averaged over all test trajectories:

$$\mathrm{nRMSE} = \frac{1}{n_{\mathrm{test}}} \sum_{i=1}^{n_{\mathrm{test}}} \left\| u_{0:T}^{(i)} - \tilde{u}_{0:T}^{(i)} \right\|_2 \bigg/ \left\| u_{0:T}^{(i)} \right\|_2 \tag{5}$$

where $u$ and $\tilde{u}$ denote the ground-truth and predicted trajectories. For coupled systems, nRMSE is computed separately for each variable, and the final error is the sum: $\text{err} = \text{nRMSE}^{\alpha} + \text{nRMSE}^{\beta}$. For all methods, reported errors are computed on normalized fields; given the scale-invariant nature of the metric, the results are expected to be similar under the original physical scaling.

## 5.2 1-DIMENSIONAL CAPACITIVELY COUPLED PLASMA FLUID MODEL

In this section, we look at a 1-dimensional capacitively coupled plasma (CCP) model as an example to demonstrate the performance of the proposed model in a parameteric setting.

The simplified CCP model describes a low-temperature plasma that is generated between two electrodes when an alternating voltage is applied, operating in a manner similar to a capacitor. In this system, the oscillating electric field accelerates electrons, which then collide with neutral gas atoms or molecules, producing ions and additional electrons that sustain the plasma. CCPs are widely used in microelectronics manufacturing, including etching and thin-film deposition for semiconductor devices, as well as in materials processing and surface engineering (Lieberman & Lichtenberg, 1994; Rauf et al., 2023).

The governing equation is defined as follows:

$$\partial_t n_e = -\partial_x \Gamma_e + R, \qquad \text{(Electron continuity equation)}$$
$$\partial_{xx} \phi = -e/\epsilon_0 (n_e - n_{io}), \quad \text{(Poisson equation)} \tag{6}$$

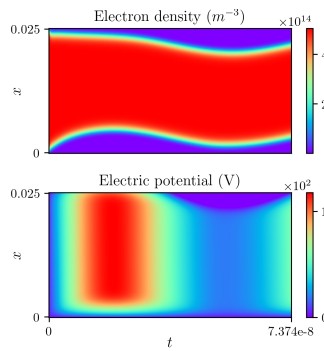

Figure 2: The spatial domain and boundary conditions.

where $n_e(x, t)$ and $\phi(x, t)$ denote electron density and electric potential, which are the solutions of the equations, and $\partial_\square$ refers to a partial derivative with respect to $\square$. $\Gamma_e$ is electron flux, defined as diffusion flux and drift flux such as $\Gamma_e = -D\partial_x n_e - \mu n_e \partial_x \phi$ with electron diffusion coefficient $D$ and electron mobility coefficient $\mu$, and ion density is defined as $n_{io} = R_0(x_2 - x_1)\sqrt{m_i/eT_e}$. $R$ and $R_0$ denote reaction rate and coefficient, respectively. The ion density is assumed constant in this model.

**Physical parameters** There are several physical parameters characterizing a domain geometry, boundary conditions (BCs), and governing physical dynamics (See Table 5 in Appendix). Among them, in this study, we consider fixed parameters for geometry (illustrated in Figure 2), but varying parameters that define the boundary conditions and the dynamics. In the BCs, the zero boundary condition is given to the electron density (at $x = 0, L$). The electric potential satisfies a Dirichlet condition (at $x = 0$), and a time-periodic Dirichlet condition at $x = L$, $V(t) = V_0 \sin(2\pi t)$. That is, the potential at the right boundary oscillates sinusoidally with amplitude $V_0$ (driving voltage) and frequency $f$ and we vary $V_0$ in the experiments. This models the externally applied radio-frequency forcing that sustains the discharge in CCP. Along with $V_0$, we consider two other physical parameters that governs the dynamics: reaction coefficient, $R_0$, and ion mass, $m_i$. Figure 3 depicts solution snapshots of $n_e(x,t)$ and $\phi(x,t)$, simulated over a single cycle, configured with $V_0 = 100$, $R_0 = 2.7 \times 10^{20}$, and $m_i = 40 \times 1.67 \times 10^{-27}$.

Figure 3: An example solution snapshot of electron density and electric potential.

### 5.2.1 DATA SETUP

**Data generation** We collect solution snapshots by numerically simulating the governing equations using a finite-difference solver. The spatial domain is discretized into 128 cells, and the time domain into 100,000 steps per cycle, with the cycle determined by the boundary driving frequency of 13.56 MHz. A fixed initial condition is used to focus on scenarios where capturing physical dynamics is critical. We then subsample every 1000th snapshot, yielding 100 temporal indices per solution. Physical parameters are sampled on an equidistant grid, which contains 100 elements: $V_0 \in [100, 300]$, $R_0 \in [2.7 \times 10^{19}, 2.7 \times 10^{20}]$, and $m_i \in [1.67 \times 10^{-26}, 6.68 \times 10^{-26}]$. This setup yields 100 trajectories, each corresponding to one parameter configuration, which are then randomly split into training and test sets with a 9:1 ratio.

### 5.2.2 NUMERICAL RESULTS

**1-dimensional parameter space**   We perform three separate experiments, each on a dataset obtained by varying a single physical parameter: $R_0$, $V_0$, or $m_i$. Table 1 summarizes the results, reporting prediction errors on the test set for the proposed methods and the baseline models. The proposed $\text{FNO}_x$ architecture ($\text{FNO}_x$, without the parametric extension) provides notable improvement over the baseline methods. The overall second best model in this set of experiments is CMWNO and, compared to its performance, $\text{FNO}_x$ achieve the prediction accuracy improved up to 38% (in the varying reaction rate scenario). The parametric extensions, $\text{pFNO}_x$ and $\text{hpFNO}_x$, further improve the performance, achieving up to 55% improvement over the best performing non-parameterized baselines. Compared to $\text{hyperFNO}$, which is also informed explicitly by the physical parameters through hypernetworks, the proposed variants, particularly, $\text{hpFNO}_x$ demonstrates significant improvements (more than 40% for all cases).

Table 1: Performance comparisons between the proposed methods and the baseline methods. All models are tested on three parameters ($R_0$, $V_0$ and $m_i$) individually. The performance is measured in the relative $\ell^2$-error (nRMSE); the reported numerical values refer to the mean ($\pm$ std. dev.).

| Model | Reaction rate | Driving voltage | Ion mass |
|---|---|---|---|
| $\text{FNO}_m$ | 0.0403 ($\pm$ 0.0012) | 0.0791 ($\pm$ 0.0243) | 0.0363 ($\pm$ 0.0067) |
| $\text{FNO}_c$ | 0.0375 ($\pm$ 0.0055) | 0.0873 ($\pm$ 0.0200) | 0.0299 ($\pm$ 0.0030) |
| CFNO | 0.0315 ($\pm$ 0.0093) | 0.0428 ($\pm$ 0.0064) | 0.0333 ($\pm$ 0.0076) |
| $\text{HyperFNO}_c$ | 0.0278 ($\pm$ 0.0117) | 0.0355 ($\pm$ 0.0160) | 0.0253 ($\pm$ 0.0088) |
| $\text{MWT}_c$ | 0.0409 ($\pm$ 0.0032) | 0.0639 ($\pm$ 0.0038) | 0.0403 ($\pm$ 0.0040) |
| CWMNO | 0.0312 ($\pm$ 0.0023) | 0.0526 ($\pm$ 0.0521) | 0.0241 ($\pm$ 0.0036) |
| $\text{DON}_c$ | 0.0844 ($\pm$ 0.0058) | 0.2147 ($\pm$ 0.0414) | 0.1035 ($\pm$ 0.0174) |
| $\text{U-NET}_c$ | 0.1084 ($\pm$ 0.0345) | 0.0844 ($\pm$ 0.0354) | 0.1719 ($\pm$ 0.1022) |
| $\text{FNO}_x$ | 0.0193 ($\pm$ 0.0059) | 0.0345 ($\pm$ 0.0108) | 0.0212 ($\pm$ 0.0062) |
| $\text{pFNO}_x$ | 0.0194 ($\pm$ 0.0075) | 0.0278 ($\pm$ 0.0053) | 0.0142 ($\pm$ 0.0021) |
| $\text{hpFNO}_x$ | **0.0154** ($\pm$ 0.0029) | **0.0192** ($\pm$ 0.0040) | **0.0128** ($\pm$ 0.0017) |

**Computational/memory aspects**   Since prediction accuracy alone does not provide a complete picture of the experimental results, Figure 4 presents additional information under the varying driving voltage scenario, including model size (4a) and computational timing (4b), alongside the models' prediction accuracy. The model size is measured in the number of trainable model parameters and the timing reports a per-epoch training time measured and averaged across the total number of epochs. The both figures provide the same observation: the proposed methods $\text{FNO}_x$, $\text{pFNO}_x$, and $\text{hpFNO}_x$ achieve improvement in accuracy without sacrificing the model size (i.e., memory footprint) and the computational wall time too much. Additionally, Figure 4c shows the test loss trajectories over the training epoch. Along with the computational timing per epoch (Figure 4b), the loss trajectories provide a complete picture on models' computational requirements in achieving the presented accuracy.

**Study on varying $T_{\text{in}}$**   Although the true dynamics depend on physical parameters, all non-parametric NOs in Table 1 achieve reasonable accuracy. This is because the NOs take as input a window of past snapshots, characterized by $T_{\text{in}}$, akin to delay embedding in dynamical systems (Takens, 2006). By consuming multiple consecutive time steps, the NOs implicitly encode information about system parameters through temporal correlations, enabling them to infer dynamics.

Thus, the benefits of using the parameterized version of $\text{FNO}_x$ can be more pronounced when $T_{\text{in}}$ becomes smaller. To see this, we extend the experimentation for varying $T_{\text{in}} = \{10, 5, 2, 1\}$.[1] Table 2 reports the results comparing the performance of $\text{FNO}_c$ and $\text{FNO}_x$ and their two parametric variants ($\text{pFNOc}$, $\text{hpFNOc}$), and ($\text{pFNO}_x$, $\text{hpFNO}_x$), respectively. As $T_{\text{in}}$ decreases, the overall model performance, which is expected. For $T_{\text{in}} = \{10, 5\}$, the improvement achieved by $\text{hpFNO}_x$ over $\text{FNO}_x$ over 20%. For $T_{\text{in}} = \{2, 1\}$, while the hp variants maintain the accuracy level around

---

[1] For $T_{\text{in}} = \{2, 1\}$, training instability was observed across some random seeds; due to this, we trained models with 10 random seeds and report the average over the five runs with the higher test prediction accuracy.

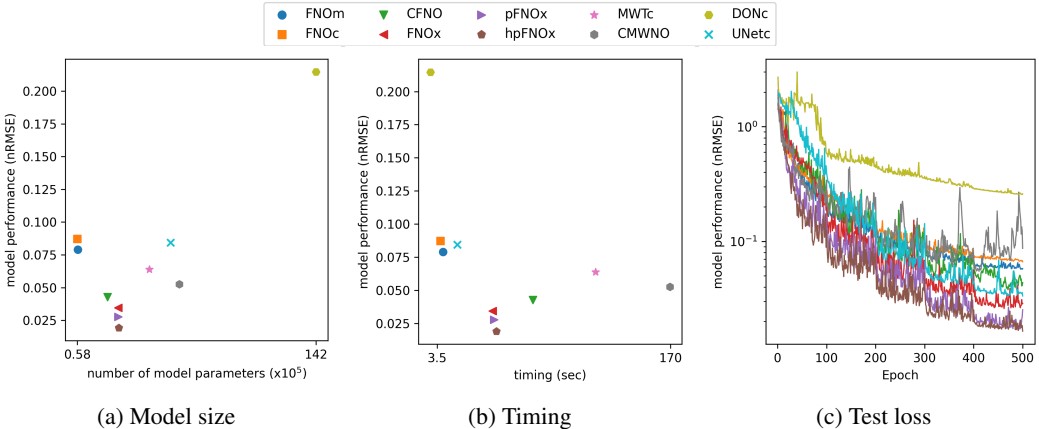

(a) Model size          (b) Timing          (c) Test loss

Figure 4: [1D CCP] Plots depict the number of model parameters versus the model performance (left), computational timing versus the model performance (middle), and the test loss trajectories over epochs (right). These results are from the varying driving voltage scenario.

13∼16% relative errors. Considering that only initial conditions are available in most practical setups, maintaining this level of performance could be considered as important step forward for FNOs.

Table 2: Performance comparisons of non-parameterized ($\texttt{FNO}_\texttt{c}$, $\texttt{FNO}_\texttt{x}$) and parameterized ($\texttt{pFNO}_\texttt{c}$, $\texttt{hpFNO}_\texttt{c}$, $\texttt{pFNO}_\texttt{x}$, $\texttt{hpFNO}_\texttt{x}$) models for varying $T_{\text{in}} = \{10, 5, 2, 1\}$. All models are tested on the reaction rate case. The performance is measured in the relative $\ell^2$-error; the reported numerical values refer to the mean ($\pm$ std. dev.). $^\dagger$ indicates that the models fail to learn the dynamics and $^\ddagger$ indicates that the model performance are averaged from the best 5 runs out of 10 total runs.

| Model | $T_{\text{in}} = 10$ | $T_{\text{in}} = 5$ | $T_{\text{in}} = 2$ | $T_{\text{in}} = 1$ |
|---|---|---|---|---|
| $\texttt{FNO}_\texttt{c}$ | 0.0375 ($\pm$0.0055) | 0.1324 ($\pm$0.0304) | 1.0048$^\dagger$ ($\pm$0.6356) | 1.4484$^\dagger$ ($\pm$0.3128) |
| $\texttt{pFNO}_\texttt{c}$ | 0.0334 ($\pm$0.0101) | 0.0923 ($\pm$0.0276) | 0.2151$^\ddagger$ ($\pm$0.0374) | 0.3757$^\ddagger$ ($\pm$0.0679) |
| $\texttt{hpFNO}_\texttt{c}$ | 0.0196 ($\pm$0.0021) | 0.0804 ($\pm$0.0237) | 0.1515$^\ddagger$ ($\pm$0.0070) | 0.1609$^\ddagger$ ($\pm$0.0043) |
| $\texttt{FNO}_\texttt{x}$ | 0.0193 ($\pm$0.0059) | 0.0406 ($\pm$0.0093) | 1.2832$^\dagger$ ($\pm$0.4298) | 1.8143$^\dagger$ ($\pm$0.0558) |
| $\texttt{pFNO}_\texttt{x}$ | 0.0194 ($\pm$0.0075) | 0.0464 ($\pm$0.0158) | 0.1640$^\ddagger$ ($\pm$0.0155) | 0.2522$^\ddagger$ ($\pm$0.1376) |
| $\texttt{hpFNO}_\texttt{x}$ | **0.0154** ($\pm$0.0029) | **0.0317** ($\pm$0.0022) | **0.1324**$^\ddagger$ ($\pm$0.0242) | **0.1372**$^\ddagger$ ($\pm$0.0100) |

## 5.3 GRAY–SCOTT EQUATIONS

As the second benchmark problem, we consider the Gray–Scott equations, which form a system of coupled reaction-diffusion equations describing the spatio-temporal dynamics of interacting chemical species. We consider a parameterized dynamics of this equation, where the parameters are diffusion coefficients and feed rate. The task of the NOs are performing predictions of time-dependent PDE solutions in the autoregressive manner, and we follow the same training/testing protocols shown in the first benchmark problem. We refer to Appendix E for details (the equations, the experimental setups, and additional results).

Table 3 summarizes the performance of the proposed and baseline models on the Gray–Scott benchmark, where the task is to predict the spatiotemporal evolution of two reacting chemical species under varying feed-rate conditions. The proposed

Table 3: Performance comparisons: The performance is measured in the relative $\ell^2$-error; the reported numerical values refer to the mean ($\pm$ standard deviation)

| Model | Performance |
|---|---|
| $\texttt{FNO}_\texttt{m}$ | 0.0116 ($\pm$ 0.0026) |
| $\texttt{FNO}_\texttt{c}$ | 0.0097 ($\pm$ 0.0030) |
| $\texttt{CFNO}$ | 0.0161 ($\pm$ 0.0045) |
| $\texttt{MWT}_\texttt{c}$ | 0.0092 ($\pm$ 0.0021) |
| $\texttt{CWMNO}$ | 0.0293 ($\pm$ 0.0078) |
| $\texttt{DON}_\texttt{c}$ | 0.0138 ($\pm$ 0.0020) |
| $\texttt{U-Net}_\texttt{c}$ | 0.0159 ($\pm$ 0.0071) |
| $\texttt{FNO}_\texttt{x}$ | 0.0075 ($\pm$ 0.0016) |
| $\texttt{pFNO}_\texttt{x}$ | 0.0041 ($\pm$ 0.0010) |
| $\texttt{hpFNO}_\texttt{x}$ | 0.0022 ($\pm$ 0.0006) |

variants consistently outperform all baseline methods, achiev-
ing relative $\ell^2$-errors of 0.0075, 0.0041, and 0.0022 for $\texttt{FNO}_x$, $\texttt{pFNO}_x$, and $\texttt{hpFNO}_x$, respectively. Among the baselines, the multiwavelet-based model ($\texttt{MWT}_c$) performs best with 0.0092 error, yet the parameterized versions of our model reduce this by more than half. These results indicate that incorporating parameter conditioning and lightweight coupling in the Fourier domain substantially improves predictive accuracy while maintaining the compactness of the original FNO architecture. Overall, the $\texttt{hpFNO}_x$ variant achieves up to 72% lower error compared to the strongest baseline, demonstrating effective generalization across parameterized reaction-diffusion dynamics.

## 6 CONCLUSION

This work has investigated extensions of Fourier neural operators (FNOs) for modeling parameterized and coupled time-dependent partial different equations. To enhance the capability of FNOs in handling parameterized dynamics, we have proposed a novel modulation-based architectural modification that incorporates parameter information in a lightweight yet effective manner. For coupled systems, we have defined design spaces of FNOs and conducted an in-depth systematic investigation to identify economic architectures that preserve efficiency while capturing inter-component interactions. In addition, we have introduced a novel benchmark problem on plasma physics, which introduces rich parameterized dynamics and contributes a valuable testbed to the community. Experimental results have demonstrated that our proposed approaches achieve significant improvements in predictive performance, while maintaining model size and runtime efficiency comparable to standard FNOs.

## 7 ETHICS STATEMENT

This study focuses on methodological advances in machine learning for computational physics problems, and we do not anticipate any significant ethical concerns arising from the proposed approaches.

## 8 REPRODUCIBILITY STATEMENT

To support reproducibility, we provide detailed descriptions of the proposed methods, including model architectures, algorithms, loss definitions, hyperparameters, training setups, and baseline comparisons. All experiments are described in the main text and appendix, and upon acceptance we will release the full source code to further facilitate reproduction of our results.

## 9 ACKNOWLEDGEMENT

The authors acknowledge funding support from Applied Materials Inc. K. Lee acknowledges partial support from the U.S. National Science Foundation under grant IIS 2338909. K. Lee also acknowledges Research Computing Jennewein et al. (2023) at Arizona State University for providing HPC resources that have contributed to the partial research results reported within this paper.

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

# A DESIGN SPACES OF FNOS

## A.1 DESCRIPTION ON EACH COMPONENT

**Data space**

Ⓓ1 Separate single-channel format. Each variable is treated as an independent function, $a(x) = (a^\alpha(x), a^\beta(x))$, and discretized separately as $\mathbf{a}^\alpha, \mathbf{a}^\beta \in \mathbb{R}^m$. This representation keeps the two components distinct throughout the architecture.

Ⓓ2 Two-channel format. The input is represented as a vector-valued function, $a : \mathcal{X} \to \mathbb{R}^2$, $x \mapsto (a^\alpha(x), a^\beta(x))$, which in a discretized form becomes $\mathbf{a} = [\mathbf{a}^\alpha, \mathbf{a}^\beta] \in \mathbb{R}^{m \times 2}$. The output is represented analogously as $u(x) = [u^\alpha(x), u^\beta(x)]^\mathsf{T} \in \mathbb{R}^2$ and the discretized form $\mathbf{u} = [\mathbf{u}^\alpha, \mathbf{u}^\beta] \in \mathbb{R}^{m \times 2}$.

**Lift operator, $P$**

Ⓟ1 The lift operator $P : \mathbb{R}^1 \to \mathbb{R}^{d_v}$ and shared across the two input components, so that both variables are mapped into the latent space using the same transformation.

Ⓟ2 Alternatively, the lift operator can consist of two separate mappings, $P = (P^\alpha, P^\beta)$, each specific to one input component. This allows the two variables to have independent latent representations from the beginning.

Ⓟ3 Compatible with Ⓓ2, the lift operator is defined as $P : \mathbb{R}^2 \to \mathbb{R}^{d_v}$, mapping a two-channel input to a hidden state.

**Fourier layers**

Ⓛ1 The point-wise linear map is a single operator shared by the both variables.

Ⓛ2 The point-wise linear map is defined as two separate operators $(W^\alpha, W^\beta)$.

Ⓖ For coupled systems, we define the global spectral convolution to perform coupling only in Fourier space. Specifically, each variable is first transformed independently, $\tilde{v}^\square(k) = \mathcal{F} v_\ell^\square(k)$, for $\square = \{\alpha, \beta\}$. Their spectral representations are then combined through a shallow encoder network $f^{\text{enc}}(\tilde{v}^\alpha(k), \tilde{v}^\beta(k))$, followed by the usual mode-wise kernel multiplication $\tilde{\tilde{v}}(k) = R_\phi(k)\tilde{v}(k)$. The result is decomposed into variable-specific coefficients using another shallow network, $(\tilde{\tilde{v}}^\alpha(k), \tilde{\tilde{v}}^\beta(k) = f^{\text{dec}}(\tilde{\tilde{v}}(k))$, and finally mapped back to the data space via the inverse Fourier transform, $v_{\ell+1}^\square(k) = \mathcal{F}^{-1}(\tilde{\tilde{v}}^\square(k))$, for $\square = \alpha, \beta$.

Ⓖ2 The standard global spectral convolution performs the Fourier transform, mode-wise kernel multiplication, followed by the inverse Fourier transform: $\mathcal{F}^{-1}\left(R_\phi \cdot (\mathcal{F} v_\ell)\right)(x)$.

**Projection operator, $Q$**

Ⓠ1 The projection operator is shared across both output components, $Q : \mathbb{R}^{d_v} \to \mathbb{R}^1$.

Ⓠ2 The projection operator is defined separately for each output component, $Q = (Q^\alpha, Q^\beta)$, with $Q^\square : \mathbb{R}^{d_v} \to \mathbb{R}^1$ for $\square \in \alpha, \beta$. From the adaptive-basis view perspective, $Q$ can differ in whether the basis and the coefficients are shared or separate across the two variables:

- Ⓠ2a Shared basis and separate coefficients, and

- Ⓠ2b Separate basis and shared coefficients, and

- Ⓠ2c Separate basis and separate coefficients.

Ⓠ3 Similar to Ⓟ3, this projection operator is defined as $Q : \mathbb{R}^{d_v} \to \mathbb{R}^2$, mapping a hidden state to a two-channel output.

## A.2 SUMMARY OF THE COMPONENTS AND RESULTING MODELS

Table 4 summarizes the architectural components and configuration options discussed in Sections 4.1-4.2 and Appendix A.1, highlighting how shared or separate design choices can be applied to handle parameterized dynamics and multi-variable (coupled) systems.

Table 4: Summary of architectural components and configurations explored in `FNOx`.

| Component | Design choice / notation | Purpose and interpretation |
|---|---|---|
| **Parameterization** | Input augmentation (`pFNO`) / Hypernetwork-based modulation (`hpFNO`) | Introduces explicit dependence on the physical parameter $\mu$. In the hypernetwork-based approach, a small network outputs layer-wise shifts $s_\ell(x, \mu)$ that modulate activations without regenerating full weights. |
| **Lift operator, $P$** | Shared ($P_1$) / Separate ($P_2$) across variables | Projects input fields to latent space. In multi-variable settings, a shared lift operator uses the same transformation for all variables, while a separate one allows variable-specific latent representations. |
| **Point-wise linear map, $W$** | Shared ($L_1$) / Separate ($L_2$) across variables | Controls local feature transformations in each Fourier layer. Sharing maintains parameter efficiency, while separating allows each variable to evolve differently in latent space. |
| **Global spectral convolution, $G$** | Standard ($G_2$) / Coupled ($G$) | Defines non-local interactions in Fourier space. The coupled variant mixes spectral representations of multiple variables via an encoder-decoder, enabling cross-variable interaction within a single operator. |
| **Projection operator, $Q$** | Shared ($Q_1$) / Separate basis and coefficients ($Q_{2a,b,c}$) | Maps latent states back to the physical output. In coupled systems, using separate bases and coefficients lets each variable form its own representation while improving expressivity. |

Now we list realizations of FNOs obtained from some combinations of the above design choices.

- $\text{FNO}_\text{m}$: P2 + P3 + L1 + G2 + Q3 - a two-channel input is processed by a standard FNO to produce a two-channel output.
- $\text{FNO}_\text{c}$: P1 + P1 + L1 + G2 + Q1 - two single-channel variables are concatenated in a discrete representation, i.e., $[\mathbf{a}^{\alpha\top}, \mathbf{a}^{\beta\top}]^\top \in \mathbb{R}^{2m}$; the concatenated input is processed by a standard FNO to produce a single-channel vertically-concatenated output, i.e., $[\mathbf{u}^{\alpha\top}, \mathbf{u}^{\beta\top}]^\top \in \mathbb{R}^{2m}$.
- $\text{FNO}_\text{x}$: This is a class of the proposed models, which can be realized with the combinations of P1 + P1/P2 + L1/L2 + G + Q1/Q2. In the experimentation, with $\text{FNO}_\text{x}$, we refer to P1 + P2 + L2 + G + Q2c unless otherwise stated.

## A.3 VISUAL COMPARISONS OF ARCHITECTURES

We provide a set of visual representations of the proposed architectures to illustrate the modifications made in this work and highlight the difference from the original FNO architecture (Figure 5).

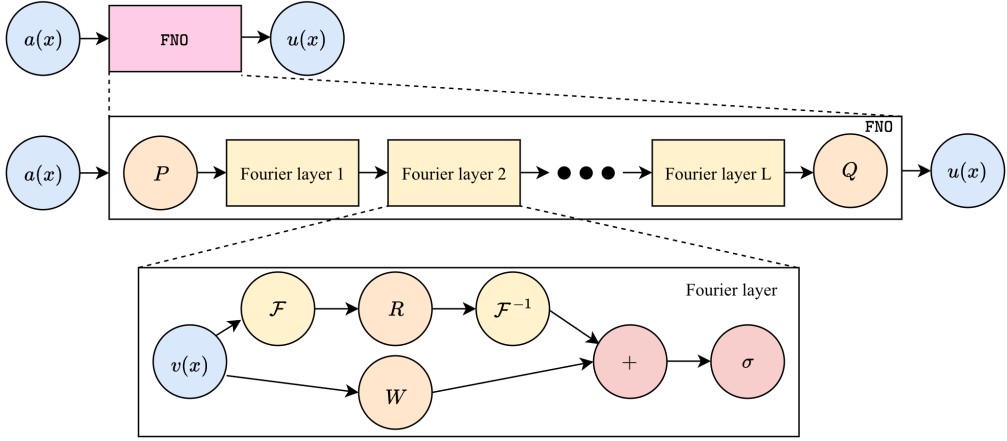

Figure 5: The original FNO architecture

**Parametric extension** In the first set of illustrations, we present the visual representations of the parameteric extension of FNOs: `pFNO` and `hpFNO`. Figure 6 depicts the `pFNO` architecture, which

augments the physical parameters with the input function, which is then used as the input to the FNO layers.

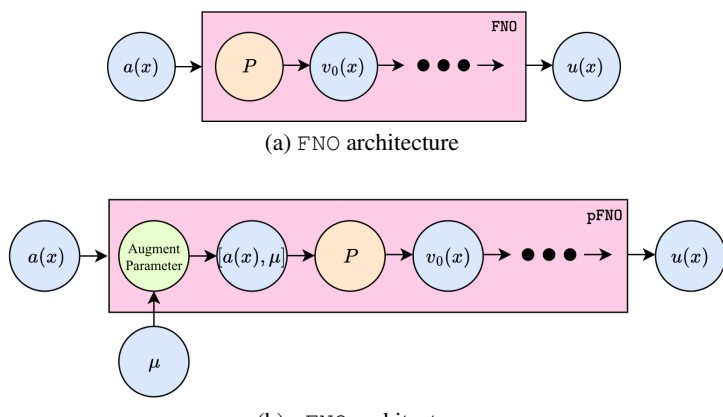

(a) `FNO` architecture

(b) `pFNO` architecture

Figure 6: The abstract representations of (a) `FNO` and (b) `pFNO` architectures are depicted.

Figure 7 illustrates the additional computational flow from the physical parameter to the shift function $s(\mu, x)$, which then acts as the bias in the Fourier layer.

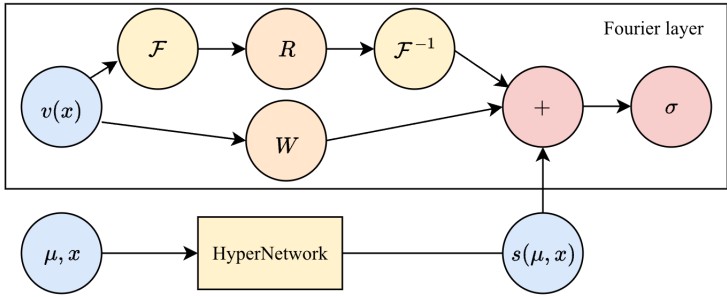

Figure 7: The proposed `hpFNO` architecture. The hypernetwork produces a parameter-dependent 'shift' term, $s(\mu, x)$, which acts as a parameter-dependent bias term in each Fourier layer.

**Coupling extension**

The next set of visual representations explains how the proposed architecture models coupled physical behavior. Figure 8 illustrates the design spaces for the lift operator $P$ and the projection operator $Q$. These two components determine how information is encoded and decoded across variables. In multi-variable systems, $P$ controls whether the input fields share the same latent representation or retain variable-specific mappings, while $Q$ defines how the learned latent features are projected back into each physical field. Figure 8b further illustrates three possible configurations of $Q$ (i.e., shared basis and coefficients, shared basis with separate coefficients), and fully separate basis and coefficients. The light orange blocks indicate shared operators, whereas the darker orange blocks indicate variable-specific operators.

Figure 9 presents a graphical overview of the proposed coupled extension of the `FNO` architecture. The coupling mechanism is introduced through an encoder-decoder pair that mixes the spectral representations of multiple variables directly in Fourier space. The encoder aggregates information from the transformed variables, and the decoder distributes the coupled representation back to each channel, enabling cross-variable interaction without substantially increasing model size. This design preserves the efficiency of the original `FNO` while introducing controlled coupling through a single shared operator.

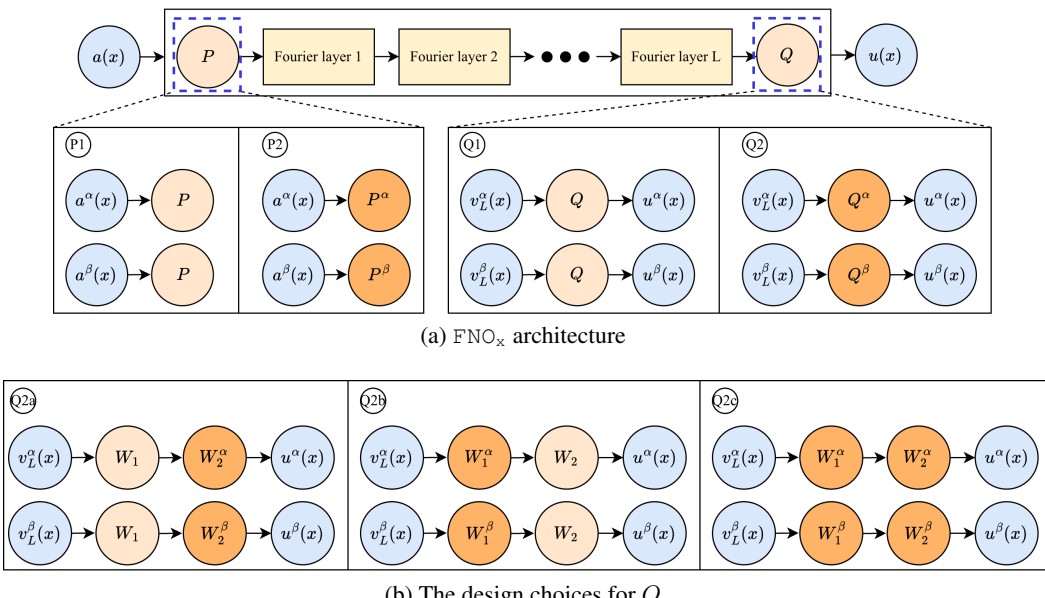

(a) $\text{FNO}_x$ architecture

(b) The design choices for $Q$

Figure 8: The standard and the proposed FNO architecture for modeling coupled behavior.

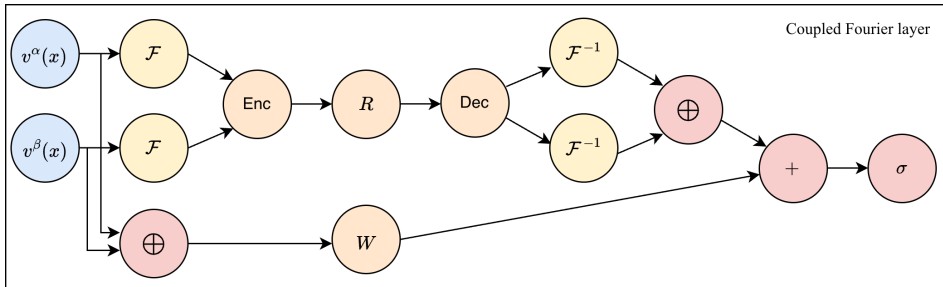

Figure 9: The proposed coupled extension of $\text{FNO}$ architecture. The $\oplus$ symbol indicates the concatenation operation.

# B    IMPLEMENTATION DETAILS

## B.1    DESCRIPTIONS AND SPECIFICATIONS OF THE BASELINES AND THE PROPOSED MODELS

**$\text{FNO}_m$ / $\text{FNO}_c$ / $\text{FNO}_x$**    For all FNO-based baselines, we consider the same hyper-parameter 4 Fourier layers with the modes and width ($k = 12$ and $d_v = 20$). $P$ is parameterized as a linear layer and $Q$ is parameterized as a shallow MLP with one hidden layer. All nonlinearity functions are ReLU (Nair & Hinton, 2010). This fixed hyperparameter setup is to control the representational capacity contributed by these hyperparameters. Consequently, any observed performance gains can be attributed to the architectural modifications introduced in each variant rather than to differences in model capacity.

For defining the $\text{FNO}_x$, we consider the combination, ⓟ1 + ⓠ2 + ⓠ2c, which employs the shared lift operator, the separate point-wise linear maps, the proposed coupled global spectral convolution layer, and the separate sets of basis functions and separate sets of coefficients in the projection operator. The encoder/decoder, $f^{\text{enc}}/f^{\text{dec}}$, in the Fourier layers are set to simple linear layers. This choice is obtained from the ablation study.

**$\text{pFNO}_c$ / $\text{pFNO}_x$**    These parameterized variant takes the standard input (i.e., the input to $\text{FNO}_c$ and $\text{FNO}_x$), but augmented with the physical parameters, $\mu$.

**hpFNO$_c$ / hpFNO$_x$**   These variants require one additional component, the hypernetwork, $f^{\text{hyper}}$. In our implementation, we consider a simple linear layer to parameterize the hypernewtork, following the design used in (Dupont et al., 2022), and we model the hypernetwork to take $(x, \mu)$ along with the discrete reprsentation of the windowed history, $\{u(t - \tau)\}_{\tau=0}^{T_{\text{in}}-1}$. Thus, the hypernetwork becomes a local transform similar to the lift operator.

**HyperFNO$_c$**   HyperFNO (Alesiani et al., 2022) integrates a base FNO with a hyper-network; the hyper-network takes the PDE parameter vector $\mu$ as input and generates FNO's weights. For each Fourier layer $\ell$, the hypernetwork generates a subset of the FNO parameters such as the pointwise linear map ($W_\ell$), the spectral convolution weights ($R_\ell$), and the lift/projection operators ($P$ and $Q$), according to

$$W_\ell = h_W(\mu; \theta_W), \quad R_\ell = h_R(\mu; \theta_R), \quad P = h_P(\mu; \theta_P), \quad Q = h_Q(\mu; \theta_Q), \tag{7}$$

where $h_\square(\mu; \theta_\square)$ for $\square = \{W, R, P, Q\}$ denotes a hyper-network with its own model parameters $\theta_\square$.

To control model size, two variants are proposed: the Addition form, which scales existing weights by $\mu$-dependent factors, and the Taylor form, which further factorizes the modulation to reduce parameter count.

In the Addition form, the hypernetwork modulates the base FNO weights through low-rank parameterization:

$$
\begin{aligned}
h_R(\theta_R^\ell, \, \mu) &= R_0^\ell + \left( V_R^\ell \operatorname{diag}(h_{R,\mu}(\mu)) \, V_R^{\ell\top} \right) \odot_{\text{row,col}} R_1^\ell, \\
h_W(\theta_W^\ell, \, \mu) &= W_0^\ell + \left( U_W^\ell \, h_{W,\mu}(\mu) \right) \odot_{\text{row}} W_1^\ell,
\end{aligned}
\tag{8}
$$

where $\odot_{\text{row}}$ and $\odot_{\text{row,col}}$ denote row-wise and element-wise modulation, respectively, and $h_{\square,\mu}(\mu)$ refers to a shallow MLP. In our experimentation, we use a two-layer MLP with width 128.

In the Taylor form, the modulation is expressed through a first-order expansion around the base parameters:

$$
h_R(\theta_R^\ell, \, \mu) = R_0^\ell \odot \left( I + V_R^\ell \operatorname{diag}(h_{R,\mu}(\mu)) \, V_R^{\ell\top} \right), \quad h_W(\theta_W^\ell, \, \mu) = W_0^\ell \odot \left( I + U_W^\ell \, h_{W,\mu}(\mu) \right),
\tag{9}
$$

where $\odot$ denotes element-wise multiplication.

In the experiments described in the main body, we consider the Taylor variant; in the Appendix, we provide the results of both variants. To differentiate those two methods, we denote them by `HyperFNO`$_c$`-T` and `HyperFNO`$_c$`-A`.

**MWT$_c$ / CMWNO**   The multiwavelet-based operator learning framework (`MWT`) has been introduced as an approach to leverage multiwavelet expansions for differential equations. By decomposing solution operators into multiwavelet bases, `MWT` captures both global structures and localized variations in a flexible and efficient way, capable of handling multi-scale dynamics and heterogeneous patterns. The coupled extension of this has been introduced in the sequel (Xiao et al., 2023), which is named as `MWT`$_c$ and is designed to take vertically concatenated input variables.

Extending this idea, the coupled multiwavelet neural operator (`CMWNO`) adapts the `MWT` framework to systems of coupled partial differential equations. `CMWNO` introduces a strategy that decouples the operator kernels across different wavelet components during decomposition and reconstruction. This enables the model to efficiently learn and represent the interactions between multiple coupled fields while keeping computation tractable. Computationally, `CMWNO` consists of two distinct operators that communicate by exchanging hidden representations. A randomized scheduling mechanism, referred to as the 'dice' algorithm, determines the order in which these operators share information, enabling flexible and efficient coordination between the coupled components.

For both `MWT` and `CMWNO`, we use the implementation available from the `Github` repository[2]. The scaling parameter $\alpha$ is set to 12, the channel multiplier $c$ is set to 16, the wavelet polynomial order $c$ is set to 4, and the base polynomial basis is set to the 'Legendre' polynomials. The dice strategy

---

[2]https://github.com/joshuaxiao98/CMWNO/

is utilized as follows: for each batch we throw a random number from a uniform distribution [0,1] and if the sampled random number is below 0.5, the second operator takes the first operator's hidden representation through the roll-out. On the opposite case, the first operator takes the first operator's hidden representation through the roll-out.

**CFNO**  CFNO mimics the operations of CMWNO, i.e., two separate operators enabling coupling effect via sharing a hidden representation of one operator to another. For CFNO, the backbone is FNO instead of CFNO. Also, the same dice approach described above is utilized.

Same as other FNO-based baselines, we consider the same hyper-parameter 4 Fourier layers with the modes and width ($k = 12$ and $d_v = 20$). $P$ is parameterized as a linear layer and $Q$ is parameterized as a shallow MLP with one hidden layer. All nonlinearity functions are ReLU. For CFNOs, we have two separate realizations of FNOs and in the third layer of CFNOs, the hidden representation from one CFNO is passed to another CFNO, where the direction is dictated by the dice approach.

**DeepONets**  For the implementation of DeepONets (DONs), we base our implementation on a publicly available PyTorch version. [3] We modify the DONs to take multivariate input and produce multivariate output. We explore the combination of a (shared/separate) trunk net and a (shared/separate) branch net, and find that two separate branch networks and a shared trunk network perform the best. Each branch network takes the same input, which is concatenated time-windowed snapshots; then one branch network produces coefficients for the electron density and another branch network produces coefficients for the electric potential. That is,

$$
n_e(x, t+1) = \sum_{i=1}^{p} f_i^{\text{branch}, n_e}(\bar{a}_{\leq t}) f_i^{\text{trunk}, n_e}(x),
$$

$$
\phi(x, t+1) = \sum_{i=1}^{p} f_i^{\text{branch}, \phi}(\bar{a}_{\leq t}) f_i^{\text{trunk}, \phi}(x)
\tag{10}
$$

where $f^{\text{branch}, n_e}$ and $f^{\text{branch}, \phi}$ denote two separate MLPs, taking time-windowed input

$$
\bar{a}_{\leq t} = \begin{bmatrix} n_e(x, t) & \cdots & n_e(x, t - T_{\text{in}} + 1) \\ \phi(x, t) & \cdots & \phi(x, t - T_{\text{in}} + 1) \end{bmatrix},
\tag{11}
$$

where $T_{\text{in}}$ decides the window length. The trunk net is modeled as a separate network, producing two separate sets of basis such that

$$
[f^{\text{trunk}, n_e}(x), f^{\text{trunk}, \phi}(x)] = [W^{n_e} f^{\text{trunk}}(x) + b^{n_e}, W^{\phi} f^{\text{trunk}}(x) + b^{n_\phi}],
\tag{12}
$$

where $f^{\text{trunk}}$ denote a shared MLP, and $(W^{n_e}, b^{n_e})$ and $(W^{\phi}, b^{n_\phi})$ denote a set of model parameters specific to the electron density and electric potential, respectively.

For each branch networks, we use 4 hidden layers with 1024 neurons in each layer. For the trunk network, we use 3 hidden layers with 1024 neurons. For both network types, we consider ReLU for nonlinear activation.

**U-Net**  For the implementation of U-Nets, we base our implementation on the code base we use for DeepONets. We modify the U-Net to take 1-dimensional and 2 channel inputs, and produce the output with the same dimensional specification. The U-Net architecture we consider has an encoder-decoder structure with skip connections.

The encoder progressively reduces the spatial resolution while increasing the feature dimension. It begins with an initial block (DoubleConv → ReLU), followed by four downsampling stages (Max-Pool1d → DoubleConv → ReLU), where each stage halves the sequence length and expands the channel dimension. The spatial resolution is halved at every downsampling and the channel dimension is updated (32→64→64→128→128).

The decoder mirrors this process with four upsampling stages (Upsample → Concat (skip-connection) → DoubleConv → ReLU). At each stage, the feature map is upsampled by a factor of two, concatenated with the corresponding encoder feature (skip connection), and refined by a double convolution block. Finally, the output layer applies 1D convolution to project the features to the desired number of output channels.

---

[3]This is accessible through https://github.com/camlab-ethz/ConvolutionalNeuralOperator.

## B.2 TRAINING DETAILS

All FNO variants and wavelet-based operators are trained for 500 epochs using the Adam optimizer (Kingma & Ba, 2015) with a learning rate of 0.0025, weight decay 0.0001, and mini-batches of size 10. For U-Nets, we consider the same hyperparameters, except for the total training epochs, which is set to 1,000, which is to mitigate the effect of the increased model sizes. For DeepONets, we again consider the same hyperparameters, except for the learning rate, which is set to 0.0001 and the total training epochs, which is set to 2,500 which is to compensate the decreased learning rate. For all baseline models, we repeat 5 different runs for varying random seeds. For all methods, we apply a linearly decaying temporal weighting to the rollout loss, emphasizing short-horizon prediction accuracy while progressively reducing the contribution of longer-horizon errors to mitigate compounding autoregressive effects.

## B.3 ORTHOGONALITY LOSS

To measure the orthogonality of the learned basis functions, $\Psi(x) = [\psi_1(x), \ldots, \psi_{n_q}(x)]$, we define a discrete inner product over the spatial domain. For a spatial domain $D$ and a measure $\rho(x)$, the continuous inner product is $\langle \psi_i, \psi_j \rangle = \int_D \psi_i(x)\psi_j(x)\mathrm{d}\rho(x)$. In practice, we approximate this integral using numerical quadrature. Specifically, we employ Gauss-type quadrature rules, which select quadrature nodes $x_k \in D$ and associated weights $w_k$, yielding $\langle \psi_i, \psi_j \rangle \approx \sum_{m=1}^M w_k \psi_i(x_k)\psi_j(x_k)$.

With this discrete formulation, we compute the Gram matrix, $[G]_{ij} = \langle \psi_i, \psi_j \rangle$, for $i, j = 1, \ldots, n_q$ and quantify deviation from orthonormality via $\mathcal{L}_{\mathrm{ortho}} = \|G - I_{n_q}\|_F^2$, leading to the ultimate loss $\mathcal{L} = \mathcal{L}_{\mathrm{mse}} + \lambda_{\mathrm{ortho}}\mathcal{L}_{\mathrm{ortho}}$. This approach allows for a principled enforcement of orthonormality in the learned basis while ensuring consistency with the underlying continuous inner product and measure. For $\lambda_{\mathrm{ortho}}$, we consider $\{0.1, 0.01, 0.001\}$.

## C    1D CCP: DESCRIPTION

### C.1    GOVERNING EQUATIONS

The simplified one-dimensional capacitively coupled plasma physics is described by two equations: electron continuity equation and Poisson equation. The electron continuity equation describes the conservation of particles; electron density changes in time because of electron flows out at the surface and because they can be created/destroyed locally due to the chemical reactions. The Poisson equation relates how the electrostatic potential varies in space to the distribution of charges (i.e., electrons and ions) in the plasma. The governing equations are defined as follows:

$$\frac{\partial n_e}{\partial t} = -\frac{\partial \Gamma_e}{\partial x} + R, \qquad \text{(Electron continuity equation)}$$
$$\partial_{xx}\phi = \frac{e}{\epsilon_0}(n_e - n_{io}), \qquad \text{(Poisson equation)} \tag{13}$$

where $n_e(x, t)$ and $\phi(x, t)$ denote electron density and electric potential, which are the solutions of the equations, and $\partial_\square$ refers to a partial derivative with respect to $\square$. $\Gamma_e$ is electron flux, defined as diffusion flux and drift flux such as

$$\Gamma_e = -D\partial_x n_e - \mu n_e \partial_x \phi \tag{14}$$

with electron diffusion coefficient $D$ and electron mobility coefficient $\mu$, which is defined as $D = eT_e/m_e\vartheta_m$. Here, $\mu = e/m_e\vartheta_m$. Here, $e$, $T_e$, $m_e$, and $\vartheta_m$ denote elementary charge, electron temperature, electron mass, and collision frequency. The ion density is defined as $n_{io} = R_0(x_2 - x_1)\sqrt{m_i/eT_e}$, where $R(x)$ and $R_0$ denote reaction rate and coefficient, respectively. The reaction rate is a spatially dependent quantity such that

$$R(x) = \begin{cases} R_0, & x \in [x_1, x_2] \cup [L - x_2, L - x_1] \\ 0, & \text{otherwise} \end{cases}. \tag{15}$$

The boundary conditions are specified as

$$\begin{aligned} n_e = 0, \phi = 0, & \qquad \text{at } x = 0, \\ n_e = 0, \phi = V(t) & \qquad \text{at } x = L, \end{aligned} \tag{16}$$

where $V(t) = V_0 \sin(2\pi t)$. That is, the zero boundary condition is given to the electron density (at $x = 0, L$). The electric potential satisfies a homogeneous Dirichlet condition (at $x = 0$), and a time-periodic Dirichlet condition at $x = L$. That is, the potential at the right boundary oscillates sinusoidally with amplitude $V_0$ and frequency $f$.

Table 5 lists up the important parameters, their description, and their values. Other important constants are elementary charge, $e = 1.6 \times 10^{-19}$ (C) and vacuum permittivity $\epsilon_0 = 8.854 \times 10^{-12}$ ($\mathrm{C^2 kg^{-1} m^{-3} s^2}$).

Table 5: Input parameter symbols, descriptions, and values

| Symbols | Descriptions | Values (unit) |
|---|---|---|
| $L$ | Domain Length | 0.025 (m) |
| $x_1, x_2$ | Reaction area | 0.005(m), 0.01(m) |
| $f$ | Driving frequency | 13.56 (MHz) |
| $V_0$ | Driving voltage amplitude | [100, 300] (V) |
| $R_0$ | Reaction rage coefficients | $[2.7 \times 10^{19}, 2.7 \times 10^{20\text{'}}]$ $(\mathrm{m^{-3} s^{-1}})$ |
| $T_\mathrm{e}$ | Electron temperature | 3 (eV) |
| $m_\mathrm{i}$ | Ion mass | $[1.67 \times 10^{-26}, 6.68 \times 10^{-26}]$ (kg) |
| $m_\mathrm{e}$ | Electron mass | $9.109 \times 10^{-31}$ (kg) |
| $\vartheta_m$ | Collision frequency | $10^8$ $(\mathrm{s^{-1}})$ |

## C.2 VISUALIZATION OF THE SOLUTION SNAPSHOTS

To provide better understanding on the dynamics of the benchmark problem, we present some collections of solution snapshots for varying input parameters, reaction rate (Figure 10) and driving voltage (Figure 11) in the first RF cycle. All Figures presents the two spatio-temporal fields, the electron density $n_\mathrm{e}(x, t)$ and the electric potential $\phi(x, t)$, on a $t - x$ plane, in a heatmap format.

Figure 10 shows the solution snapshots for varying reaction rates $R_0 = \{2.7 \times 10^{19}, 1.485 \times 10^{20}, 2.7 \times 10^{20}\} m^{-3} s^{-1}$. The reaction rate serves as a quantity that determines how fast ionization occurs. The reaction rate drives the plasma density and sustainment in a CCP discharge. The higher the reaction rates, the faster the ionization is, and the higher plasma density is. Reaction rate can be controlled using power in plasma processing systems. Figure 10 shows some example solution snapshots for varying reaction rate; from the left panel to the right panel, the reaction rate increases, which results in higher electron density and thinner sheaths.

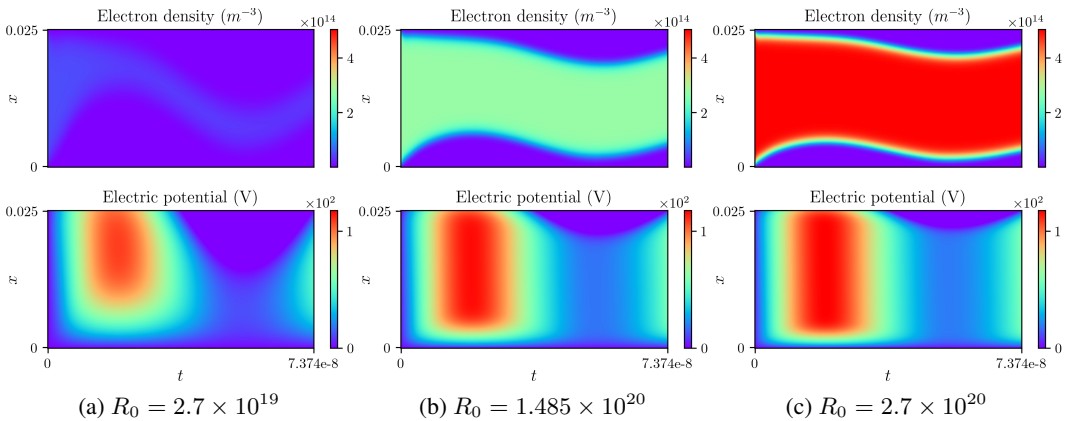

(a) $R_0 = 2.7 \times 10^{19}$     (b) $R_0 = 1.485 \times 10^{20}$     (c) $R_0 = 2.7 \times 10^{20}$

Figure 10: [1D CCP] Solution snapshots presented in heatmap; the solutions are collected for varying reaction rates $R_0 = \{2.7 \times 10^{19}, 1.485 \times 10^{20}, 2.7 \times 10^{20}\}$.

Figure 11 shows the solution snapshots for varying driving voltages $V_0 = \{100, 200, 300\}$ in the first RF cycle. The driving voltage, given as the boundary condition, establishes the time-varying electric potential fields. The higher the driving voltage, the larger is the sheath voltage, resulting in stronger electric fields. The sheath is thicker at higher electric field. Figure 11 shows example solution snapshots for varying driving voltage: (a) $V_0 = 100$, (b) $V_0 = 200$, and (c) $V_0 = 300$.

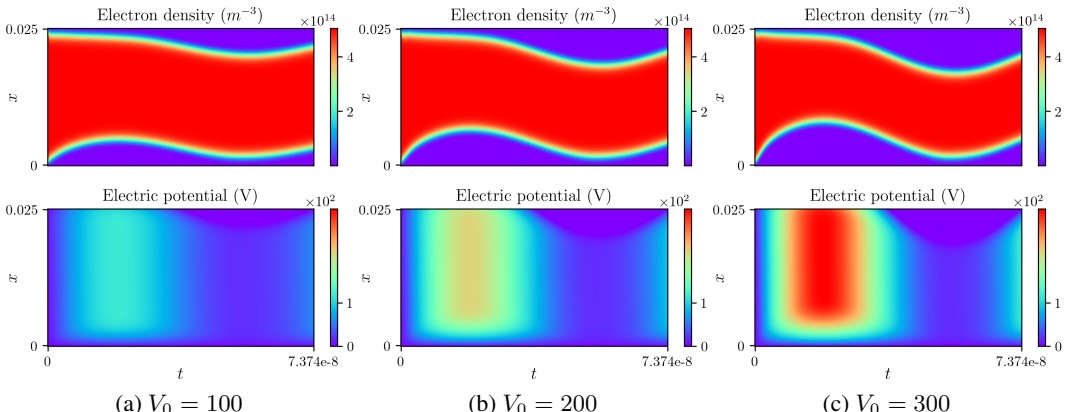

(a) $V_0 = 100$ (b) $V_0 = 200$ (c) $V_0 = 300$

Figure 11: [1D CCP] Solution snapshots presented in heatmap; the solutions are collected for varying driving voltages $V_0 = \{100, 200, 300\}$.

Likewise, varying the ion-mass also provides variations in the dynamics. However, compared to other two physical parameters, the reaction rate and the driving voltage, changing the ion-mass parameter has slightly less physical impact on the dynamics and thus, the solution snapshots.

## D  1D CCP: ADDITIONAL EXPERIMENTAL DETAILS AND RESULTS

### D.1  1D CCP RESULTS

In the following sections, we present additional experimental results that are not presented in the main body due to the space limit. The additional experiments include an ablation study on different design choices of FNOs, a study on effectiveness of the proposed hypernetwork and modulation based architecture for the parametric extension, and a study on extrapolation in the parameter space.

#### D.1.1  ADDITIONAL RESULTS

Continuing from the experimental results presented in the main text (i.e., the "1-dimensional parameter space" experimentation), we present the further detailed information regarding the computations: the model sizes and the computational timings, depicted in Figure 12. This time, the information obtained from the experimentation with the datasets that vary the reaction rate and the ion masses. Again, the model size is measured in the number of trainable model parameters and the timing reports a per-epoch training time measured and averaged across the total number of epochs. We can make the similar observations here: the proposed methods $\text{FNO}_x$, $\text{pFNO}_x$, and $\text{hpFNO}_x$ achieve improvement in accuracy without sacrificing the model size and the computational wall time. In case of the ion-mass scenario, we see relatively small improvements compared to other two scenarios (the reaction rates and the driving voltage). This is largely due to the fact that varying the ion-mass provide less changing physical dynamics and taking the windowed history input, $\{u(t-\tau)\}_{\tau=0}^{T_{in}-1}$, allows other non-parameterized NOs to infer those physical parameters to some extent. Still, the proposed methods $\text{FNO}_x$, $\text{pFNO}_x$, and $\text{hpFNO}_x$ improves the performance in that scenario while minimally affecting the model size and the training time.

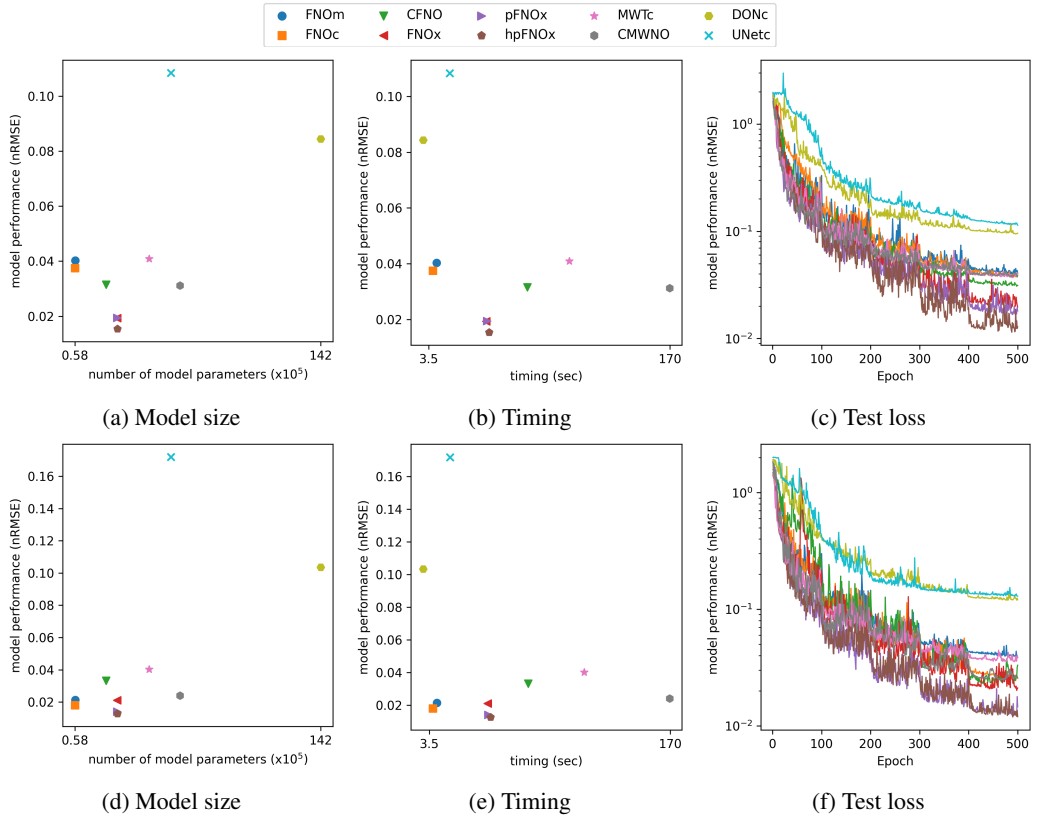

Figure 12: [1D CCP] Plots depict the number of model parameters versus the model performance (left), computational timing versus the model performance (right), for varying reaction rates $R_0$ (top) and ion masses $m_i$ (bottom)

### D.1.2 COUPLED EXTENSION

**Ablation on model architectures** To investigate the effectiveness of each design component, we perform an ablation study on several different combinations of the design choices. We begin with the *default* combination, (P1) + (L1) + (Q1), which consists of the shared lift/projection operators and the point-wise linear map in the Fourier layers. We first consider replacing the point-wise linear map in the Fourier layer to be a set of separate point-wise linear maps, $W^\alpha$ and $W^\beta$. After that, we investigate the projection operator by testing three different combinations: Shared/separate basis functions $\Psi$ and shared/separate coefficients $\Xi$. Finally, we test the effect of a set of separate lift operators. Table 6 reports the results of this ablation study. The normalization shown in the last

Table 6: Performance comparisons of different realizations of FNO$_x$. The models are trained for 1-dimensional parameter space, (reaction rate). The performance is measured in the relative $\ell^2$-error (nRMSE); the reported numerical values refer to the mean ($\pm$ std. dev).

| Variant | Description | Relative errors |
|---|---|---|
| (P1) + (L1) + (Q1) | Default setup | 0.0341 ($\pm$ 0.0044) |
| (P1) + (L2) + (Q1) | Separate $W$ | 0.0281 ($\pm$ 0.0048) |
| (P1) + (L2) + (Q2a) | Separate $W$ and coefficients $\Xi$ | 0.0259 ($\pm$ 0.0032) |
| (P1) + (L2) + (Q2b) | Separate $W$ and basis $\Psi$ | 0.0315 ($\pm$ 0.0028) |
| (P1) + (L2) + (Q2c) | Separate $W$ and basis $\Psi$ and coefficients $\Xi$ | 0.0275 ($\pm$ 0.0062) |
| (P1) + (L2) + (Q2b) w. norm | Separate $W$ and coefficients $\Xi$ with layer norm | 0.0205 ($\pm$ 0.0043) |
| (P1) + (L2) + (Q2c) w. norm | Separate $W$ and basis $\Psi$ and coefficients $\Xi$ with layer norm | 0.0193 ($\pm$ 0.0059) |
| (P2) + (L2) + (Q2c) w. norm | Separate $P$ and $W$ and basis $\Psi$ and coefficients $\Xi$ with layer norm | 0.0204 ($\pm$ 0.0035) |

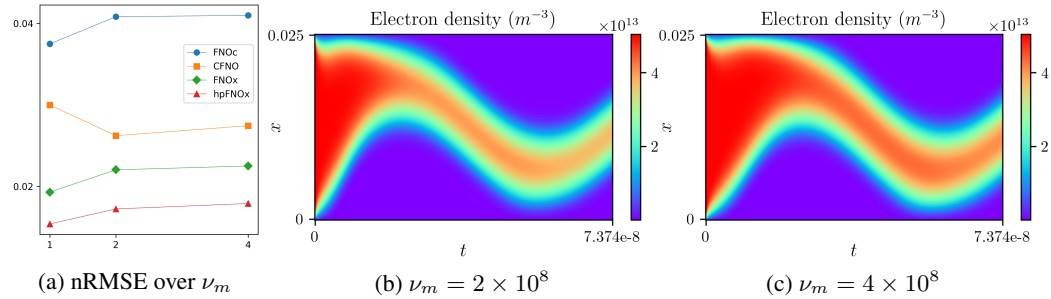

(a) nRMSE over $\nu_m$       (b) $\nu_m = 2 \times 10^8$       (c) $\nu_m = 4 \times 10^8$

Figure 13: [1D CCP] (a) The model performance measured in nRMSE across varying $\nu_m = \{1, 2, 4\} \times 10^8$. (b) and (c): Solution snapshots collected for varying $\nu_m = \{2 \times 10^8, 4 \times 10^8\}$ while keeping all other parameter values.

entries of the table is layer norm applied to the pre-activation values in the projection operator (i.e., $W_1 v_T(x) + b_1$). From this ablation study, we choose the combination, ⓟ1 + Ⓛ2 + ⓠ2c with layer norm, as $\mathtt{FNO_x}$ for all experimentation.

**Varying collisional coupling** To examine the effect of coupling strength in the 1D CCP system, we vary the electron–neutral collision frequency parameter $\nu_m$ across $1 \times 10^8$, $2 \times 10^8$, and $4 \times 10^8$. As summarized in Figure 13a, we observe a general trend of increasing nRMSE with higher $\nu_m$ across all models. While the proposed $\mathtt{hpFNO_x}$ exhibits a mild degradation at the largest $\nu_m$, it still achieves the lowest overall errors (nRMSE $\approx 0.015$–$0.018$).

This behavior is consistent with the underlying plasma physics: as the system becomes more collisional, electron transport is increasingly suppressed. In particular, increasing $\nu_m$ strengthens the coupling between charged species and the background gas through enhanced momentum exchange, while simultaneously reducing both the diffusion coefficient $D = eT_e/(m_e \nu_m)$ and the mobility $\mu = e/(m_e \nu_m)$. The reduced transport leads to stiffer solution snapshots with sharper spatial gradients and weaker temporal smoothness (Figures 13b–13c). Consequently, the resulting fields exhibit steeper localized structures that are more challenging for spectral operator networks to approximate.

Taken together, these results suggest that higher-collisional regimes introduce both stronger coupling and increased stiffness, leading to moderately higher errors. We note, however, that it is difficult to fully isolate the influence of coupling strength alone, since changes in $\nu_m$ simultaneously affect both the degree of coupling and the stiffness of the governing dynamics, thereby altering the learning difficulty and the numerical conditioning of the operator.

### D.1.3 PARAMETRIC EXTENSION

**Study on the effectiveness of the proposed model architecture** To further investigate the effectiveness of the proposed parameteric extension, we compare the performances of the non-parameterized versions, $\mathtt{FNO_c}$ and $\mathtt{FNO_x}$ against their own two parameterized versions, ($\mathtt{pFNO_c}$, $\mathtt{hpFNO_c}$) and ($\mathtt{pFNO_x}$, $\mathtt{hpFNO_x}$).

Table 7 reports the models' performance measured in all three physical parameter scenarios. Taking the non-parameterized version as the baseline, Table 7 reports the achieved prediction accuracy of each model and their improvements over the non-parameterized versions, presented in the percentage. The results indicate that simply augmenting physical parameters to input provide some improvements in prediction accuracy. However, the new architectural modification (i.e., hyper-network and modulation) brings significantly improved results (upto 68.01% improvement). As a reference point, we also report the performance of two variants of $\mathtt{HyperFNO_c}$-T and -A (Taylor and Addition) as well. The overall performance of $\mathtt{HyperFNO_c}$ are in-between those of $\mathtt{pFNO_c}$ and $\mathtt{hpFNO_c}$, suggesting that the parameter-efficient modulation technique (i.e., latent shift modulation) provides benefits in the given experimental contexts.

**Study on the out-of-distribution samples** In the main experiments, we follow the standard FNO testing strategy. In our parameterized scenarios, however, we can explicitly sample parameters that

Table 7: Performance comparisons of non-parameterized ($\texttt{FNO}_c$, $\texttt{FNO}_x$) and parameterized ($\texttt{pFNO}_c$, $\texttt{hpFNO}_c$, $\texttt{pFNO}_x$, $\texttt{hpFNO}_x$) models. All models are tested on three scalar parameters (reaction rate, driving voltage, and ion mass) individually. The performance is measured in the relative $\ell^2$-error; the reported numerical values refer to the mean ($\pm$ standard deviation, improvement over non-parameterized version).

| Model | Reaction rate | Driving voltage | Ion mass |
|---|---|---|---|
| $\texttt{FNO}_c$ | 0.0375 ($\pm$ 0.0055) | 0.0873 ($\pm$ 0.0200) | 0.0299 ($\pm$ 0.0030) |
| $\texttt{pFNO}_c$ | 0.0334 ($\pm$ 0.0101, 10.94 %) | 0.0454 ($\pm$ 0.0094, 47.92 %) | 0.0284 ($\pm$ 0.0023, 5.08 %) |
| $\texttt{hpFNO}_c$ | 0.0196 ($\pm$ 0.0021, 47.57 %) | 0.0279 ($\pm$ 0.0016, 68.01 %) | 0.0210 ($\pm$ 0.0036, 29.84 %) |
| $\texttt{FNO}_x$ | 0.0193 ($\pm$ 0.0059) | 0.0345 ($\pm$ 0.0108) | 0.0212 ($\pm$ 0.0062) |
| $\texttt{pFNO}_x$ | 0.0194 ($\pm$ 0.0075, -0.51 %) | 0.0278 ($\pm$ 0.0053, 19.30 %) | 0.0142 ($\pm$ 0.0021, 32.91 %) |
| $\texttt{hpFNO}_x$ | **0.0154** ($\pm$ 0.0029, 20.12 %) | **0.0192** ($\pm$ 0.0040, 44.40 %) | **0.0128** ($\pm$ 0.0017, 39.83 %) |
| $\texttt{HyperFNO}_c\texttt{-T}$ | 0.0278 ($\pm$ 0.0117) | 0.0355 ($\pm$ 0.0160) | 0.0253 ($\pm$ 0.0088) |
| $\texttt{HyperFNO}_c\texttt{-A}$ | 0.0276 ($\pm$ 0.0119) | 0.0401 ($\pm$ 0.0143) | 0.0250 ($\pm$ 0.0098) |

lie entirely outside the training range, which we define as out-of-distribution (OOD) testing. As a benchmark case, we vary the reaction rate $R_0$. For training, 101 equidistance values of the reaction rate is sampled from $[2.7\times10^{19}, 2.7\times10^{20}]$; for OOD testing, the interval is expanded outward by $0.2 \times 10^{19}$ in both directions. Table 8 reports the performance of all considered models on ID test samples and OOD test samples. Overall, the performance measured on OOD produces increased numbers, but the trend across the compared methods remains the same. The proposed $\texttt{FNO}_x$ model family produce more accurate predictions than the existing methods.

Figure 14 depicts the per-parameter nRMSE measured for the reaction coefficients that are sampled in the OOD region. As the index increases (from 1 to 10, sampled uniformly from $[2.7\times10^{20}, 2.9\times10^{20}]$), the parameter is farther away from the training distribution. All considered methods show a gradual increase in nRMSE as the chosen parameter departs from the training range. The parameterized versions ($\texttt{pFNO}_c$, $\texttt{hpFNO}_c$) and ($\texttt{pFNO}_x$, $\texttt{hpFNO}_x$) achieve lower errors compared to their non-parameterized counterparts, indicating improved extrapolation capability. This trend is expected for data-driven surrogate models, and the smooth growth of nRMSE suggests that the proposed parameterized extensions remain stable even when evaluated beyond the training domain. We note that this analysis is intended to investigate general extrapolation behavior rather than to claim superior OOD performance compared to large-scale or foundation models (e.g., (Rahman et al., 2024; Herde et al., 2024; Holzschuh et al., 2025)).

Table 8: Performance comparisons of non-parameterized ($\texttt{FNOc}$, $\texttt{FNO}_x$) and parameterized ($\texttt{pFNOc}$, $\texttt{hpFNOc}$, $\texttt{pFNO}_x$, $\texttt{hpFNO}_x$) models for OOD samples (reaction rate). The performance is measured in the relative $\ell^2$-error; the reported numerical values refer to the mean ($\pm$ standard deviation, improvement over non-parameterized version).

| Model | ID | OOD |
|---|---|---|
| $\texttt{FNO}_m$ | 0.0403 ($\pm$ 0.0012) | 0.0628 ($\pm$ 0.0075) |
| $\texttt{FNO}_c$ | 0.0375 ($\pm$ 0.0055) | 0.0673 ($\pm$ 0.0097) |
| $\texttt{pFNO}_c$ | 0.0334 ($\pm$ 0.0101) | 0.0498 ($\pm$ 0.0069) |
| $\texttt{hpFNO}_c$ | 0.0196 ($\pm$ 0.0021) | 0.0374 ($\pm$ 0.0051) |
| $\texttt{FNO}_x$ | 0.0193 ($\pm$ 0.0059) | 0.0446 ($\pm$ 0.0094) |
| $\texttt{pFNO}_x$ | 0.0194 ($\pm$ 0.0075) | 0.0343 ($\pm$ 0.0059) |
| $\texttt{hpFNO}_x$ | **0.0154** ($\pm$ 0.0029) | **0.0303** ($\pm$ 0.0050) |

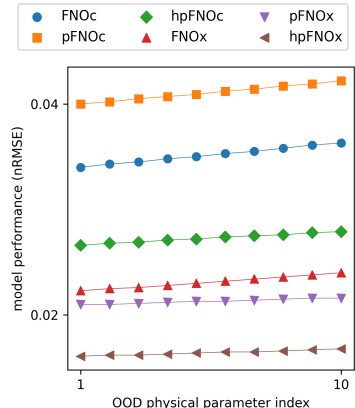

Figure 14: nRMSE over the physical parameter indices.

**2-dimensional parameter space**  Next we extend our investigation to 2D input parameter space, a parameter space spanned by the reaction rate and the driving voltage. For input parameter sampling, we consider the same ranges for each parameter $V_0 \in [100, 300]$, $R_0 \in [2.7 \times 10^{19}, 2.7 \times 10^{20}]$, construct a uniform 2D mesh grid (21×21), and randomly split them into training and test sets with 9:1 ratio. We can make observations that are similar with the ones from the 1D input parameter cases; the proposed architecture `FNO`$_x$ provides improvement in prediction accuracy and parameterized versions of `FNO`$_c$ and `FNO`$_x$ further reduce the prediction errors, leading to the best performance with `hpFNO`$_x$. The model size and the computational timing follow the similar trends with the 1D case, only except the slightly increased per-epoch computation time due to the increased number of data instances in the dataset for all methods.

Table 9: Performance comparisons for the 2D input parameter case.

| Model | Performance |
|---|---|
| FNO$_m$ | 0.0285 ($\pm$ 0.0028) |
| FNO$_c$ | 0.0252 ($\pm$ 0.0026) |
| CFNO | 0.0223 ($\pm$ 0.0023) |
| MWT$_c$ | 0.0305 ($\pm$ 0.0028) |
| CMWNO | 0.0298 ($\pm$ 0.0023) |
| DON$_c$ | 0.0409 ($\pm$ 0.0077) |
| U-Net$_c$ | 0.0461 ($\pm$ 0.0160) |
| FNO$_x$ | 0.0163 ($\pm$ 0.0009) |
| pFNO$_x$ | 0.0140 ($\pm$ 0.0012) |
| hpFNO$_x$ | **0.0124** ($\pm$ 0.0011) |

**Adaptive basis view point**  In the next set of experiments, we investigate further on the adaptive basis viewpoint given on the last projection operator $Q$. We first vary the width (that is, the number of basis functions) of the last layer of $Q$ for $n_q = \{8, 16, 32, 64, 128\}$ and measure the prediction performance of `hpFNO`$_x$ on the varying reaction rate scenario. Taking the $d_{\text{basis}} = 8$ as a baseline, Figure 15 reports the percentage of improvements (shown in the bottom filled markers). Next, we test the idea of having (close to) orthogonal basis functions and repeat the same experiments. An orthonormality condition imposed to the learned basis, $\Psi(a(x))$; we could achieve further improvement (shown in the top filled markers). As $d_{\text{basis}}$ becomes small, the effect of the orthogonality enforcement becomes higher, leading to $\sim$10% improvements. We define the orthogonality of the learned basis functions and its computation in Appendix B.3.

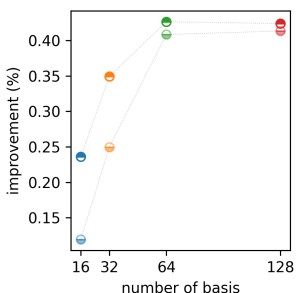

Figure 15: Num. of bases v.s. performance improvement.

**Study on hyper-network configurations**  To assess the impact of hypernetwork design, we test several configurations varying in depth and width. Specifically, we consider hypernetworks with 1 and 2 hidden layers (width 256) and 1 hidden layer (width 128), which yield nRMSE values in the range of 0.0171-0.0179 compared to 0.0154 for the linear baseline. Deeper or narrower configurations, such as a 3-layer network (width 256, nRMSE = 0.0200) or 3-layer network (width 32, nRMSE = 0.0230), show a mild degradation in performance. These results indicate that `hpFNO`$_x$ performance is relatively insensitive to moderate variations in hypernetwork architecture, while deeper or overly compact designs may introduce minor instability. The linear configuration remains the most efficient and accurate choice.

**Comparison of modulation techniques**  In addition to the proposed shift (bias) modulation, we also evaluate variants that modulate both the layer weights and biases based on the physical parameters. We denote this approach by `weight` modulation. The first variant is the weight modulation following the formulation suggested in the HyperFNO paper:

$$h_W(\theta_W^\ell, \mu) = W_0^\ell + \left( U_W^\ell \, h_{W,\mu}(\mu) \right) \odot_{\text{row}} W_1^\ell, \tag{17}$$

where $h_{W,\mu}(\mu)$ denotes the MLP-encoded parameter vector. This configuration allows both $W_\ell(\mu)$ and $s_\ell(x, \mu)$ to vary with the physical parameter $\mu$.

For the next variant, we augment the additive shift modulation with a multiplicative gating mechanism, which is in a similar spirit with Feature-wise linear modulation (FiLM) (Perez et al., 2018). We denote this approach by `scale` modulation. With this augmentation, the Fourier layer becomes:

$$v_{\ell+1}(x) = \sigma\Big(\gamma_\ell(x, \mu)\left[W \, v_\ell(x) + \left(\mathcal{K}(a; \phi) \, v_\ell\right)(x)\right] + s_\ell(x, \mu)\Big), \qquad \forall x \in D. \tag{18}$$

where the scale factor, $\gamma(x, \mu) = [\gamma_1(x, \mu), \ldots, \gamma_L(x, \mu)]$, is inferred from a hypernetwork, along with the shift $s(x, \mu)$.

Table 10: Performance comparisons of different modulation techniques applied to `hpFNO`$_c$.

|  | Reaction term | Driving voltage | Ion mass |
|---:|:---:|:---:|:---:|
| Shift only | 0.0154 ($\pm$ 0.0029) | 0.0279 ($\pm$ 0.0016) | **0.0128** ($\pm$ 0.0017) |
| `Weight` + shift | 0.0157 ($\pm$ 0.0028) | 0.0214 ($\pm$ 0.0056) | 0.0134 ($\pm$ 0.0036) |
| `Scale` + shift | 0.5037 ($\pm$ 0.0144) | 0.0279 ($\pm$ 0.0636) | 0.8550 ($\pm$ 1.8748) |
| `Scale`-0.1 + shift | 0.0151 ($\pm$ 0.0036) | **0.0170** ($\pm$ 0.0029) | 0.0137 ($\pm$ 0.0059) |
| `Scale`-0.5 + shift | **0.0146** ($\pm$ 0.0022) | 0.0180 ($\pm$ 0.0031) | 0.0157 ($\pm$ 0.0053) |

Compared to using a shift term alone, the inclusion of $\gamma_\ell(x, \mu)$ makes the modulation strictly more expressive, as it enables the network to multiplicatively amplify or suppress the latent responses in a parameter-dependent manner. However, multiplicative gating can introduce training instabilities if $\gamma_\ell$ varies too aggressively across the parameter space. To mitigate this, we employ a stabilization strategy by parameterizing the gate as

$$\gamma_\ell(x, \mu) = 1 + \eta \tanh(\tilde{\gamma}_\ell(x, \mu)), \tag{19}$$

which keeps the modulation centered near identity at initialization while still allowing smooth, learnable deviations driven by the physical parameter. Here, $\tilde{\gamma}$ becomes the hypernetwork output and $\eta$ is the hyper-parameter to decide the amplitude of deviation. We denote this variant by `scale`-$\eta$, where the suffix indicates the chosen value of $\eta$.

Table 10 provides summarized results. Overall, the amplitude-based multiplicative gating generally offers performance improvements, but it introduces several considerations. First, naive multiplicative modulation is unstable, since the unconstrained scaling factors can deviate excessively and cause large errors during training. Second, the stabilized formulation proposed in this work, which applies a scaled `tanh` function to control the amplitude of the gate, significantly improves robustness but requires searching over the hyperparameter $\eta$ to determine the appropriate scaling range. Third, the benefits of multiplicative gating are not universal across all settings; for example, in the ion mass benchmark, the configuration with the shift modulation alone achieves the best accuracy. These observations indicate that although multiplicative gating is more expressive in principle and can yield improved performance, its practical effectiveness depends on careful amplitude control and the characteristics of the specific parameterized problems.

In summary, a reasonably strong and stable parameterization can be obtained using the shift modulation alone, which delivers reliable performance across the benchmarks considered in this work. However, the amplitude based multiplicative gating provides additional expressive capacity, and with a more refined search over its scaling hyperparameters, there is room for achieving higher accuracy. This suggests a practical tradeoff: the shift modulation is simple and robust, whereas the amplitude based modulation can offer further performance gains when carefully tuned.

## E  GRAY–SCOTT EQUATIONS

The Gray–Scott equations form a system of coupled reaction-diffusion equations that describe the spatiotemporal dynamics of interacting chemical species. This model can reproduce a rich variety of natural patterns, including structure resembling bacterial colonies, spiral waves, and coral-like formations. In the system, two state variables $u$ and $v$ each undergo independent diffusion and linear growth/decay, while their interaction is governed by the nonlinear reaction term.

The governing equations are defined as follows:

$$\begin{aligned} \frac{\partial u}{\partial t} &= \epsilon_1 \frac{\partial^2 u}{\partial x^2} + F(1 - u) - \lambda_1 u v^2, \\ \frac{\partial v}{\partial t} &= \epsilon_2 \frac{\partial^2 v}{\partial x^2} - (K + F)v + \lambda_2 u v^2, \end{aligned} \tag{20}$$

where $u(x, t)$ denotes concentration of the feed species, which is continuously supplied to the system and $v(x, t)$ denotes concentration of the reactant species, which is produced and consumed through interaction with $u$. The diffusion coefficients of $u$ and $v$ are denoted as $\epsilon_1$ and $\epsilon_2$, respectively. Other

parameters include: $F$, the feed rate, representing the rate at which species $u$ is introduced into the system from an external source, $K$, he decay rate of species $v$, which accounts for natural removal, and $\lambda$, the reaction rate constant associated with the nonlinear term $-\lambda u v^2$, which describes the autocatalytic process where one unit of $u$ reacts with two units of $v$.

### E.1 EXPERIMENTATION

In the experiment, we consider the parameterized dynamics realized by varying the diffusion coefficient $\epsilon_1 \in [0.1, 10]$ and varying the feed rate $F \in [0.1, 10]$. We collect 101 samples of equi-distanced coefficient/feed rate values while the other parameters are fixed as $\epsilon_2 = 0.05$, $K = 0.1$ and $(\lambda_1, \lambda_2) = (2, 5)$. For sampling an initial condition, we follow the approach described in (Xiao et al., 2023), sampling $u(x, 0)$ using the smooth random functions in the Chebfun package (Driscoll et al., 2014) and sampling $v(x, 0)$ from Gaussian random field, $v(x, 0) \sim \mathcal{N}(0, 7^4(-2\pi + 7^2 I)^{-2.75})$. For collecting trajectories for constructing training/test datasets, initial value problems are numerically solved using a fourth-order time-integrator, exponential time differencing fourth-order Runge–Kutta (ETDRK4), utilizing the chebfun software. We base the implementation the publicly available Matlab script[4], which is the implementaion accompanyed by the research work (Xiao et al., 2023). The IVPs are solved with a spatial resolution, 1024, and then the collected solution snapshots are subsampled in the spatial domain, leading to the final spatial resolution, 128. The time integration is performed for the time interval [0,0.1] with 31 time steps.

Table 11: Performance comparisons between the proposed methods and the baseline methods. All models are tested on a scalar parameter: the diffusion coefficient. The performance is measured in the relative $\ell^2$-error; the reported numerical values refer to the mean ($\pm$ standard deviation)

| Model | Diffusion coeff, $\epsilon_1$ | Feed rate, $F$ |
|---|---|---|
| FNO$_m$ | 0.0129 ($\pm$ 0.0032) | 0.0116 ($\pm$ 0.0026) |
| FNO$_c$ | 0.0089 ($\pm$ 0.0052) | 0.0097 ($\pm$ 0.0030) |
| CFNO | 0.0093 ($\pm$ 0.0016) | 0.0161 ($\pm$ 0.0045) |
| MWT$_c$ | 0.0055 ($\pm$ 0.0011) | 0.0092 ($\pm$ 0.0021) |
| CWMNO | 0.0048 ($\pm$ 0.0015) | 0.0293 ($\pm$ 0.0078) |
| DON$_c$ | 0.0243 ($\pm$ 0.0029) | 0.0138 ($\pm$ 0.0020) |
| U-Net$_c$ | 0.0239 ($\pm$ 0.0160) | 0.0159 ($\pm$ 0.0071) |
| FNO$_x$ | 0.0039 ($\pm$ 0.0002) | 0.0075 ($\pm$ 0.0016) |
| pFNO$_x$ | 0.0027 ($\pm$ 0.0008) | 0.0041 ($\pm$ 0.0010) |
| hpFNO$_x$ | 0.0022 ($\pm$ 0.0010) | 0.0022 ($\pm$ 0.0006) |

**Prediction accuracy** Table 11 reports the prediction accuracy of all considered method. The results provide similar observations with those of the 1D CCP experimentation; the proposed methods, the parameterized variants of FNO$_x$ achieve the lowest errors compared to existing baselines with 54% (for the diffusion coefficient case) and 72% (for the feed rate case) improvements. The coupled extension of FNO, FNO$_x$, improves the performance compared to other baselines. This is achieved without significantly scarifying the training/inference time and the memory footprint compared to FNO$_m$ or FNO$_c$ while other baselines require significantly larger-sized models and/or computational times. In the varying feed rate scenario, the approaches leveraging two separate operators struggle further to capture the accurate dynamics, resulting in higher prediction errors.

**Computational/memory aspects** Figure 16 further provides more information on computing. The figure presents the model sizes (i.e., memory requirements) and the computational timing for per epoch operation measured in seconds. The first row presents the results for the diffusion coefficients and the bottom row presents the results for the feed rate. Again the proposed variants pFNO$_x$ and hpFNO$_x$ serve as an efficient (in both memory and computation) and effective surrogate for modeling the parameterized Gray–Scott equations. Along with these information, the third column

---

[4]This script is available in https://github.com/joshuaxiao98/CMWNO/.

of Figure 16 presents the test loss achieved over the course of the training. The proposed hpFNO$_x$ demonstrates faster convergence to more accurate predictions.

(a) Model size for $\epsilon_1$       (b) Timing for $\epsilon_1$       (c) Test loss for $\epsilon_1$

(d) Model size for $F$       (e) Timing for $F$       (f) Test loss for $F$

Figure 16: [GS] The results of the experiments for varying diffusion coefficients (top) and varying feed rates (bottom). Plots depict the number of model parameters versus the model performance (left), computational timing versus the model performance (middle), and the test loss trajectories over epochs (right).

**Varying spatial resolutions** We further conduct additional experiments on the Gray–Scott benchmark of varying diffusion coefficients using datasets generated at three different spatial resolutions: 64, 128, and 256 grid points. All models are trained separately on data from each resolution under identical training configurations and evaluated on corresponding test sets. The results, summarized in Figure 17, show that all considered methods maintain comparable accuracy across resolutions, consistent with the resolution-invariant nature of neural operator formulations. Among them, FNOx consistently achieves the lowest relative $\ell^2$-errors (approximately $3.3 \times 10^{-3}$ across all grid sizes). These results confirm that the proposed architecture preserves accuracy under varying spatial discretizations while maintaining the efficiency of operator learning.

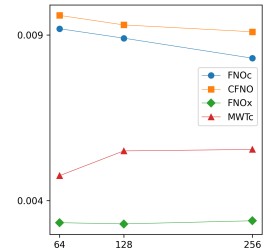

Figure 17: [GS] nRMSE over varying resolutions

## F    2D CHEMICALLY REACTING FLOW

As a next benchmark, we consider the two-dimensional chemically reacting flow of a premixed $H_2$-air flame at constant and uniform pressure Buffoni & Willcox (2010), which is modeled as a coupled system of four PDEs. Each equation governs the temporal and spatial evolution of one of the four thermo-chemical state variables, temperature and the mass fractions of $H_2$, $O_2$, and $H_2O$, which are nonlinearly coupled through diffusion, convection, and reaction terms. The governing equations for

the system is expressed as

$$\frac{\partial \boldsymbol{w}(\boldsymbol{x},t;\boldsymbol{\mu})}{\partial t} = \nabla \cdot \big(\nu \nabla \boldsymbol{w}(\boldsymbol{x},t;\boldsymbol{\mu})\big) - \boldsymbol{v} \cdot \nabla \boldsymbol{w}(\boldsymbol{x},t;\boldsymbol{\mu}) + \boldsymbol{q}\big(\boldsymbol{w}(\boldsymbol{x},t;\boldsymbol{\mu});\boldsymbol{\mu}\big), \qquad (21)$$

where $\boldsymbol{w} = [w_T, w_{H_2}, w_{O_2}, w_{H_2O}]^\mathsf{T}$ represents the state vector of the four coupled fields.

On the right-hand side, the first term captures molecular diffusion with diffusivity $\nu = 2\,\text{cm}^2\text{/s}$, the second term represents advection by a constant, divergence-free velocity field $\boldsymbol{v} = [50\,\text{cm/s}, 0]^\mathsf{T}$, and the third term $\boldsymbol{q}$ accounts for nonlinear chemical reactions coupling all species. The system is therefore strongly coupled: the evolution of each variable depends on the others through $\boldsymbol{q}$ and, to a lesser degree, through diffusive transport.

The reaction source term follows an Arrhenius-type law:

$$\boldsymbol{q}(\boldsymbol{w};\boldsymbol{\mu}) = [q_T(\boldsymbol{w};\boldsymbol{\mu}), q_{H_2}(\boldsymbol{w};\boldsymbol{\mu}), q_{O_2}(\boldsymbol{w};\boldsymbol{\mu}), q_{H_2O}(\boldsymbol{w};\boldsymbol{\mu})]^\mathsf{T}, \qquad (22)$$

where

$$q_T(\boldsymbol{w};\boldsymbol{\mu}) = Q\, q_{H_2O}(\boldsymbol{w};\boldsymbol{\mu}),$$
$$q_i(\boldsymbol{w};\boldsymbol{\mu}) = -v_i \left(\frac{W_i}{\rho}\right) \left(\frac{\rho w_{H_2}}{W_{H_2}}\right)^{v_{H_2}} \left(\frac{\rho w_{O_2}}{W_{O_2}}\right)^{v_{O_2}} A\, e^{-\frac{E}{R w_T}}, \qquad (23)$$

for $i \in \{H_2, O_2, H_2O\}$.

The stoichiometric coefficients are $(v_{H_2}, v_{O_2}, v_{H_2O}) = (2, 1, -2)$, and the molecular weights are $(W_{H_2}, W_{O_2}, W_{H_2O}) = (2.016, 31.9, 18)$ g/mol. The mixture density is $\rho = 1.39 \times 10^{-3}$ g/cm$^3$, the universal gas constant is $R = 8.314$ J/(mol·K), and the heat of reaction is $Q = 9800$ K.

The system depends on two physical parameters, $\boldsymbol{\mu} = (\mu_1, \mu_2) = (A, E)$, corresponding to the pre-exponential factor and activation energy, respectively. Hence, this problem represents a four-variable coupled nonlinear PDE system with two input parameters, capturing the essential thermo-chemical coupling of a reacting $H_2$-air mixture.

Figure 18 illustrates the computational geometry and the corresponding boundary conditions, defined as follows:

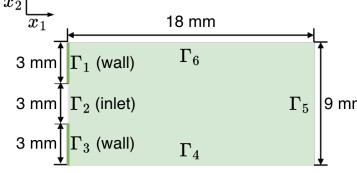

- $\Gamma_2$: inflow boundary with Dirichlet conditions specified by $T = 950$ K and $(w_{H_2}, w_{O_2}, w_{H_2O}) = (0.0282, 0.2259, 0)$,

- $\Gamma_1$ and $\Gamma_3$: boundaries with Dirichlet conditions $T = 300$ K and $(w_{H_2}, w_{O_2}, w_{H_2O}) = (0, 0, 0)$,

- $\Gamma_4$, $\Gamma_5$, and $\Gamma_6$: boundaries with homogeneous Neumann conditions,

Figure 18: [ADR] The spatial domain and boundary conditions.

and the initial condition is set to $T = 300$ K and $(w_{H_2}, w_{O_2}, w_{H_2O}) = (0, 0, 0)$, indicating that the domain initially contains no chemical species.

## F.1 EXPERIMENTATION

The numerical data are generated using a finite-difference scheme on a uniform grid of $64 \times 32$ points (i.e., $4 \times 64 \times 32$, 4 variables). For training and testing NOs, we subsample the horizontal direction to 32, leading to $32 \times 32$ for four variables. Time integration employs the second-order backward differentiation formula (BDF2) with a constant time step $\Delta t = 10^{-4}$ s and a final simulation time of $t = 0.02$ s (corresponding to $n_t = 200$ time steps). For training and testing NOs, we subsample every 5th snapshot, resulting in 40 temporal indices for each trajectory.

The two model parameters, the pre-exponential factor $A$ and the activation energy $E$, are sampled uniformly from the rectangular parameter domain $A, E) \in [2.3375 \times 10^{12}, 6.5 \times 10^{12}] \times [5.625 \times 10^3, 9 \times 10^3]$. For data collection, we construct an $8 \times 8$ uniform grid over this domain, resulting in 64 trajectories, which are then split into 56:8 (train:test).

Figure 19 summarizes the performance and efficiency of the proposed and baseline models. In this experiment, we consider a subset of the baselines considered in other experiments, namely FNO$_c$,

$\texttt{MWT}_\texttt{C}$, and $\texttt{DON}_\texttt{C}$. Particularly, $\texttt{CFNO}$ and $\texttt{CMWNO}$ are excluded from this comparison, as their architectures are primarily demonstrated for two-variable systems and it is hard to find their extension to higher-order coupling is not explicitly discussed in the original formulations. In the results, we observe that the proposed variants, $\texttt{FNO}_\texttt{x}$ and $\texttt{hpFNO}_\texttt{x}$, achieve the lowest prediction errors while maintaining competitive model size and computational cost. Relative to $\texttt{FNO}_\texttt{C}$, $\texttt{FNO}_\texttt{x}$ improves accuracy by about 41%, and $\texttt{hpFNO}_\texttt{x}$ further reduces the error by approximately 61%. Compared to the multiwavelet baseline $\texttt{MWT}_\texttt{C}$, $\texttt{hpFNO}_\texttt{x}$ still provides around 41% lower error while remaining lightweight. $\texttt{DON}_\texttt{C}$ yields nRMSE values roughly an order of magnitude larger than the other models and is therefore omitted from Figure 19 for visual clarity. Overall, the results demonstrate that spectral coupling and parameter conditioning enhance predictive accuracy and efficiency without compromising the compact structure of the original FNO architecture.

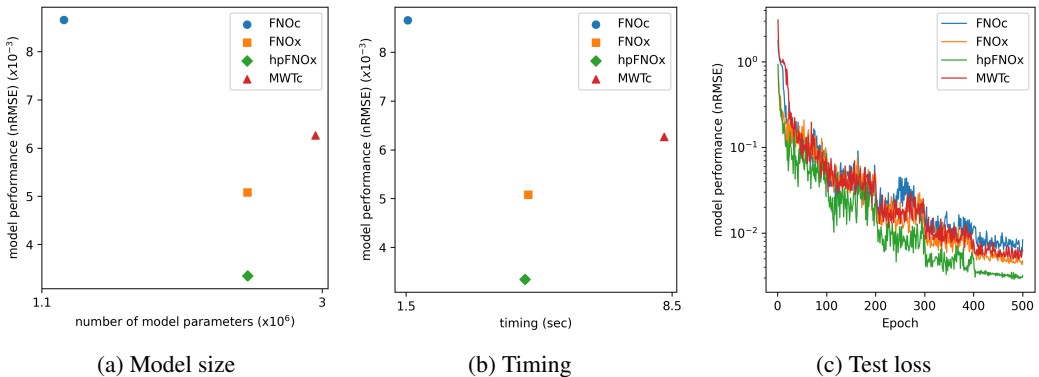

(a) Model size            (b) Timing            (c) Test loss

Figure 19: [ADR] The results of the experiments for varying pre-exponential factor and the activation energy $(A, E)$. Plots depict the number of model parameters versus the model performance (left), computational timing versus the model performance (middle), and the test loss trajectories over epochs (right).

