# OpenReview forum: "Extending Fourier Neural Operators for Modeling Parameterized and Coupled PDEs"
_ICLR.cc/2026/Conference — ICLR 2026 Poster_

### Official Review · Reviewer_UVnJ · 2025-10-24

**Soundness:** 3
**Presentation:** 3
**Contribution:** 2
**Rating:** 4
**Confidence:** 4

**Summary:**

This paper proposes two architectural extensions to Fourier Neural Operators (FNOs) to better handle two important classes of PDEs: (i) parameterized dynamics with fixed initial conditions, and (ii) systems of coupled PDEs with interacting physical fields. For parameterized dynamics, the authors introduce a lightweight hypernetwork that modulates FNO layers based on physical parameters. For coupled PDEs, they perform a systematic architectural design study to determine where and how to introduce cross-variable coupling, ultimately proposing a Fourier-space interaction scheme. The final model, FNOx (and its parametric variants pFNOx, hpFNOx), achieves significant gains over strong baselines — including coupled multiwavelet neural operators — on both a newly proposed 1D capacitively coupled plasma benchmark and the standard Gray–Scott system.

**Strengths:**

* **Addresses underexplored settings**: The paper targets two highly practical yet understudied scenarios in neural operator learning — parameterized dynamics with fixed initial conditions, and multiple interacting PDE fields. Both are common in engineering and physics simulations.

* **Effective, minimal extensions**: The proposed improvements are simple but well-motivated. The hypernetwork-based modulation (hpFNOx) conditions model behavior across parameters without blowing up model size, and the spectral-domain coupling design elegantly leverages FNO’s strengths.

* **Systematic architectural study**: The paper carefully explores different configurations (shared vs separate layers, coupling placements) and justifies its final FNOx architecture based on empirical and structural considerations.

* **Strong empirical results**: On both benchmarks, the proposed models achieve large performance gains — often reducing error by 50–70% relative to the best baseline — while maintaining similar runtime and parameter count.

* **New benchmark dataset**: The 1D plasma physics setup with tunable parameters offers a meaningful and realistic testbed for studying parametric and coupled operator learning.

* **Clarity and reproducibility**: The paper is generally well-written, with detailed experimental setups and promised code release. It adheres to good reproducibility practices.

**Weaknesses:**

* **Incremental novelty**: The core ideas — hypernetwork modulation and coupling in neural operators — are grounded in existing techniques. HyperFNO and related works already explored hypernetwork-based parameter conditioning, and Fourier-space coupling is a natural extension within FNOs.

* **Narrow scope**: The benchmarks, while appropriate, are relatively small-scale and low-dimensional (1D/2D). It’s unclear how well the approach generalizes to larger or more complex systems (e.g. 3D Navier–Stokes, multiple coupled fields).

* **Limited analysis of generalization**: There’s little discussion on how well hpFNOx handles out-of-distribution parameters or extrapolation, which is a practical concern in real applications of parameterized PDE solvers.

* **Brief treatment of second benchmark**: The Gray–Scott results are reported only briefly in the main paper, which weakens the case for generality.

**Questions:**

1. **How does the hypernetwork-modulated FNO behave on out-of-distribution (OOD) physical parameters?** Would performance degrade sharply or smoothly? Have you tested its extrapolation limits?

2. **Why is the coupling introduced only in the Fourier domain?** Have you explored combining it with local (real-space) cross-variable interactions, and how would that affect performance or efficiency?

3. **Can your proposed architecture scale to PDE systems with more than two coupled fields?** If not, what limitations would arise — architectural, memory, or training-related?

4. **How would the design choices in FNOx (e.g., Q2c, L2, G) transfer to more challenging PDE domains like turbulent 2D/3D fluid flow or real-world inverse problems?** Do you expect the same configurations to hold?

5. **To what extent does the performance improvement come from architectural tuning versus true modeling of coupled/parameterized dynamics?** Could stronger baselines (e.g. HyperFNO, CoDA-NO) close the gap?

6. **In the case of high-dimensional parameter spaces (e.g., 10–20 physical parameters), would the hypernetwork remain effective without overfitting or becoming too large?** How would you scale it?

7. **How sensitive is the performance to the size and architecture of the hypernetwork?** Could a different modulation scheme (e.g., multiplicative gating) outperform simple bias shifts?

---

> ### Author Response · Authors · 2025-11-21
> **Author rebuttal (1/3)**
>
> We thank the reviewer for the detailed and thoughtful evaluation of our work. We appreciate the recognition of the paper’s focus on underexplored yet practical settings, namely parameterized dynamics with fixed initial conditions and coupled PDE systems. We are glad that the reviewer found the proposed extensions, hypernetwork-based modulation for parameterized dynamics and Fourier-space coupling for interacting fields, to be effective and well-motivated.
>
> We also thank the reviewer for acknowledging the systematic architectural study, the strength of the empirical results, and the introduction of the new 1D plasma benchmark. We appreciate the positive feedback on the paper’s clarity, reproducibility, and adherence to good experimental practices.
>
> **Weaknesses**
>
> [W1] Incremental novelty: hyperFNO and spectral coupling
>
> We thank the reviewer for this observation. We acknowledge that the HyperFNO paper was omitted in our initial submission, and we have added the reference in the revised version. While both approaches employ hypernetworks for parameter conditioning, our method differs in design and scope. Specifically, we introduce a lightweight shift-modulation mechanism that conditions intermediate activations rather than regenerating full layer weights, which leads to lower parameter cost and stable training. In addition, our study extends the scope beyond parameterized dynamics to systematically investigate coupling mechanisms between interacting variables within a unified framework.
>
> Regarding the Fourier-space coupling, our approach mixes information directly in the spectral domain through a single shared operator, which has a different flavor from methods such as CFNO and CMWNO that exchange hidden representations between multiple operator instances. This design maintains the compactness of the base FNO while enabling efficient cross-variable interaction. Our focus is on providing a systematic and efficient architectural extension rather than proposing an entirely new operator formulation.
>
> [W2] Additional benchmark
>
> We are working to provide additional results before the author-reviewer discussion deadline.
>
> [W3/Q1] OOD test
>
> We thank the reviewer for this question. As discussed in paragraph “Study on the out-of-distribution samples,” in Appendix, we have examined this behavior to some extent and, in the revision, expand the analysis by adding visualizations showing model performance as $\mu$ moves away from the in-distribution (training) range.
>
> In general, we believe that extrapolation behavior is largely problem-dependent/parameter-dependent. In the tested reaction-coefficient scenario, all methods show a gradual increase in nRMSE as $\mu$ deviates from the training range, which is expected and consistent with typical data-driven surrogate models. The objective of our approach is to make this degradation less pronounced while keeping the model lightweight and easily deployable across different scenarios.
>
> [W4] Treatment of the second benchmark
>
> We thank the reviewer for this comment. In the revised version, we have restructured the main body to include the detailed Gray--Scott results and discussion, which were previously summarized in the appendix. The new section now presents the comparison table interpretation of the results. In addition, we added a new analysis in the appendix that discusses computational and memory efficiency and a new experiment studying model behavior under varying spatial resolutions. Hope these additions strengthen the completeness of the Gray--Scott study.
>
> **Questions**
>
> [Q1] OOD test
>
> Please refer to the response to [W3].

---

> > ### Author Response · Authors · 2025-11-21
> > **Author rebuttal (2/3)**
> >
> > [Q2] Coupling in the Fourier domain
> >
> > We thank the reviewer for this question. Coupling in the Fourier domain is chosen to maintain efficiency and leverage the spectral representation’s ability to capture global correlations between variables. The design enables lightweight cross-variable interaction through a single operator instance, where spectral representations are mixed using a compact encoder–decoder module. This approach keeps the kernel dimensionality and computational cost comparable to that of a standard FNO while still enabling effective coupling.
> >
> > A potential way to combine spectral and local (real-space) coupling would be through localized operator formulations (e.g., [Liu-Schiaffini, et al, "neural operators with localized integral and differential kernels", ICML, 2024]), which introduce spatially local kernels to capture fine-grained interactions. While such hybrid coupling could enhance expressivity, it would also require additional architectural components and higher computational cost, and is left as future work.
> >
> > For context, prior works on coupled or multiphysics neural operators (e.g., CFNO, CMWNO, CODA-NO) achieve coupling by employing multiple operator instances and exchanging hidden representations. These methods effectively capture cross-variable dependencies but result in larger models and higher memory overhead. Our single-operator design offers a more compact and scalable alternative while retaining strong predictive performance.
> >
> >
> > [Q3] Scalability w.r.t. the problem dimension and the number of variables
> >
> > We thank the reviewer for this question. Existing coupled neural operator methods such as CMWNO and CFNO generally increase the number of operator instances with the number of coupled variables. Each variable is handled by a separate operator stream, and information is exchanged between them through hidden representations. As far as we know, these approaches have not proposed mechanisms for efficiently mixing more than two variables, which would introduce both architectural and memory scalability challenges.
> >
> > In contrast, our design keeps a single FNO instance and introduces coupling through a lightweight encoder–decoder operating in the Fourier domain. This keeps the kernel tensor size identical to that of a single-variable FNO, which is important for maintaining efficiency since the kernel scales as a 3D or 4D tensor in higher dimensions. The encoder–decoder adds modest parameter overhead but does not change the spectral kernel dimensionality.
> >
> > For 2D and 3D PDEs, the scaling behavior is similar to that of standard FNOs, where kernel dimensionality increases with spatial dimension. Both our method and CFNO scale similarly in this regard. Finally, our focus is on developing a lightweight and easily deployable operator architecture.
> >
> > [Q4] The design choices in FNOx for turbulent/complex/real-world problems?
> >
> > We thank the reviewer for this question. As neural operator architectures can be viewed as learning a set of basis functions and corresponding coefficients (DeepONet, as an example, that explicitly models these two components with separate networks). In general, using (Q2c - separate bases and coefficients for each variable) is strictly more expressive than sharing them across variables, as it allows each field to construct its own functional representation. However, in complex problems, this flexibility could potentially introduce a more difficult learning problem because both bases and coefficients must be learned simultaneously, which can lead to instability or slower convergence. As a partial remedy, as shown in our experiment, imposing orthogonality or regularization on the learned bases is expected to help improve stability.
> >
> > For complex PDEs such as turbulent 2D/3D flows or real-world inverse problems, these challenges are not specific to our design but apply broadly to neural operator approaches, including FNO-based models. Such systems involve strong nonlinearity and multi-scale dynamics, which are difficult for global spectral formulations to capture effectively. For such problems, the suggested FNOx configuration would serve as a reasonable starting point, but the main performance bottleneck is likely to lie in capturing local interactions and micro-scale phenomena. Combining spectral representations with localized operator formulations or adaptive basis mechanisms could help overcome these limitations and represents a promising direction for future research.

---

> > > ### Author Response · Authors · 2025-11-21
> > > **Author rebuttal (3/3)**
> > >
> > > [Q5] Performance gap and stronger baselines
> > >
> > > We thank the reviewer for this thoughtful question. We acknowledge that disentangling architectural effects from modeling capability is inherently difficult. Our work aims to design efficient operator architectures suited for practical engineering use cases where computational resources and data availability are limited. High-fidelity neural operators, often equipped with attention mechanisms or large-scale pretraining, are more appropriate for problems involving highly complex or turbulent dynamics and can deliver stronger performance when training data and compute budgets permit. In contrast, our goal is to explore systematic and efficient architectural principles that maintain strong predictive accuracy in moderate-scale, application-driven settings.
> > >
> > > Regarding stronger baselines, HyperFNO was not designed for coupled systems but was included in our revised experiments for completeness. It performed better than the input-concatenation baseline (pFNOc) but did not outperform our proposed hpFNOc while requiring more than ten times the parameters, since it infers nearly all components of the network.
> > >
> > > CoDA-NO represents a complementary direction emphasizing large-scale pretraining and attention-based architecture expansion. The BERT-like pretraining strategy adopted in CoDA-NO is promising for building general-purpose, high-fidelity operators. In contrast, our focus is on efficient models that can be trained from scratch for specific tasks, providing a flexible alternative in settings where large-scale pretraining or heavy attention models may not be practical.
> > >
> > > [Q6] High-dim input parameter scenario
> > >
> > > We thank the reviewer for this question. The proposed hypernetwork naturally extends to higher-dimensional parameter inputs, where the increase in dimensionality mainly affects the size of the input layer while keeping the rest of the architecture unchanged.
> > >
> > > The more significant challenge in such cases would be data coverage. Consider uniform sampling; uniform sampling of a high-dimensional parameter space quickly becomes infeasible due to the curse of dimensionality, 10 points in each dimension makes 10^10 and 10^20 (10 or 20 dim space).
> > >
> > > Assuming sufficient data availability, one possible direction is to adopt large-scale pretraining and transfer-learning strategies, such as those demonstrated by [Li et al., "Towards Foundation Models for Scientific Machine Learning", NeurIPS 2023], where a model is first trained on a broad family of parameterized PDEs and then fine-tuned to specific systems.
> > >
> > > [Q7] Sensitivity to Hypernetwork or different modulation techniques
> > >
> > > *Hypernetwork*
> > > We thank the reviewer for this valuable question. In the original implementation, we followed the design used in [Dupont et al. "From Data to Functa ...", ICML 2022], where the hypernetwork is implemented as a single linear layer that maps the physical parameter vector to layer-wise shift vectors.
> > >
> > > We conducted additional experiments exploring alternative hypernetwork configurations with varying depth and width. Specifically, we tested 1- and 2-layer networks (width 256) and a 1-layer network (width 128), which produced nRMSE values in the range of 0.0171–0.0179, compared to 0.0154 for the linear baseline. Deeper or narrower configurations, such as a 3-layer network with width 256 (nRMSE: 0.0200) and width 32 (nRMSE: 0.0230), resulted in mild performance degradation.
> > >
> > > These results indicate that hpFNO performance is relatively insensitive to moderate architectural variations in the hypernetwork, while deeper or overly compact designs slightly affect stability. Overall, the single linear-layer configuration remains the most efficient and reliable choice.
> > >
> > >
> > > *Modulation*
> > >
> > > Given the reviewer 6Lqg's suggestion, we conducted a study on a different modulation technique. While parameterizing $W_\ell(\mu)$ is a possible alternative, we find that modulating activations through additive shifts provides a more efficient and stable mechanism. Shift modulation introduces fewer additional parameters, preserves the shared spectral kernels of the base FNO, and maintains smoother optimization behavior.
> > >
> > > Empirically, we implement a variant that combines parameter-dependent weight modulation following the formulation suggested in the HyperFNO paper (please refer to Eq. (17) in the revised PDF) along with the shift modulation.
> > >
> > > We test this design on the reaction rate benchmark and do not observe improvement with the weight-modulation variant (nRMSE: 0.0157) compared to the shift-modulation design (nRMSE: 0.0154). Given the similar performance and higher parameter cost, we retain the additive shift formulation for its simplicity and stability.

---

> > > > ### Author Response · Authors · 2025-11-21
> > > > **Concluding remark**
> > > >
> > > > We thank the reviewer again for the thoughtful and encouraging evaluation. We appreciate the recognition of our focus on practical, underexplored problems and the effectiveness of the proposed extensions. In the revision, we added comparisons with HyperFNO, expanded the Gray--Scott benchmark, and included analyses on scalability, OOD behavior, and hypernetwork sensitivity. We hope that these additions clarify novelty, improve readability, and strengthen the overall technical contribution.

---

> > > > > ### Author Response · Authors · 2025-11-25
> > > > > **Additional results**
> > > > >
> > > > > [W2] Additional benchmark
> > > > >
> > > > > We again thank the reviewer for this valuable comment. We have now included a new 2D chemically reacting flow problem (Appendix F) to further demonstrate the generality of the proposed approach beyond what was originally covered. This problem involves four coupled physical variables: temperature, pressure, and two reacting species. These variables interact through nonlinear advection, diffusion, and reaction processes, introducing strong coupling and a more realistic and challenging test case.
> > > > >
> > > > > The new results, presented in Appendix F (Figure 18), show that the proposed models (FNOx and hpFNOx) consistently outperform baseline methods in both accuracy and computational efficiency. These findings indicate that the proposed extensions generalize effectively to higher-dimensional, multi-variable PDE systems, extending their demonstrated applicability.

---

> ### Comment · Reviewer_UVnJ · 2025-11-26
>
> Thank you for the detailed and thoughtful revisions. The additional benchmarks, expanded analyses, and clearer discussion of novelty addressed the main questions I raised. The paper now feels noticeably stronger and more complete. With these improvements, I’m increasing my score.

---

> > ### Author Response · Authors · 2025-11-26
> > **Thank you!**
> >
> > We thank the reviewer for the thoughtful reassessment and for raising the score. We are glad that the revisions addressed the concerns. We believe that the detailed comments from the reviewer helped us strengthen the paper.

---

### Official Review · Reviewer_6Lqg · 2025-10-28

**Soundness:** 3
**Presentation:** 3
**Contribution:** 3
**Rating:** 6
**Confidence:** 3

**Summary:**

This paper extends Fourier Neural Operators (FNOs) to handle parameterized and coupled PDEs. The authors propose two lightweight modifications: (1) conditioning FNOs on physical parameters through either input concatenation (pFNO) or a small hypernetwork modulation (hpFNO), and (2) enabling coupling between multiple PDE variables by performing cross-variable mixing in Fourier space. The resulting framework (FNOx, pFNOx, hpFNOx) retains the efficiency of standard FNOs while improving accuracy for systems with varying parameters or interacting fields. The authors introduce a new benchmark based on a 1-D capacitively coupled plasma model and also evaluate on the Gray–Scott equations. The proposed hpFNO achieves up to 55–72 % error reduction over strong baselines.

**Strengths:**

- Addresses a timely and important gap: FNOs are increasingly popular for PDE surrogates, yet parameterization and coupling remain under-explored.

- The architecture modifications are simple and principled—hypernetwork modulation and Fourier-space coupling can be integrated into existing FNOs with minimal effort.

- The approach is general and modular, applicable to multidimensional parameter fields and different neural-operator families.

- Demonstrates comparisons across multiple strong baselines (FNO, CFNO, MWT, DeepONet, U-Net) with solid quantitative gains.

- Introduces a new plasma-physics benchmark that could be useful to the community.

**Weaknesses:**

1. Writing is meandering and often unfocused; core ideas and notation could be stated more directly.

2. Organization: the Related-Work section appears late; it would be clearer to position it before the Methods.

3. The Gray–Scott example feels secondary, lacking detailed analysis or ablation—appears appended to strengthen results. In fact, the experiments are limited and cannot demonstrate general applicability. The paper should be enhanced with more experiments from different PDEs.

4. The claimed generality for “parameterized and coupled PDEs” is convincing for 1-D cases but not well demonstrated on higher-dimensional or more complex systems.

5. Limited qualitative insight—plots focus on error metrics. The authors should delve deeper into the interpretation of learned coupling or modulation effects.

6. typos like "learable". should have numbered the equations.

**Questions:**

1. How sensitive is the hpFNO performance to the design of the hypernetwork (depth, parameterization, modulation type)?

2. Could the same effect be achieved by allowing \( W_\ell(\mu) \) (parameter-dependent local maps) instead of additive \( s_\ell(x, \mu) \) biases?

3. Are there stability or extrapolation limitations when \( \mu \) lies outside the training range?

4. How does coupling only in Fourier space compare to coupling in both spatial and spectral domains?

5. Can the authors comment on scalability to 2-D/3-D PDEs and memory growth with the number of coupled variables?

---

> ### Author Response · Authors · 2025-11-21
> **Author rebuttal (1/2)**
>
> We thank the reviewer for the positive and detailed evaluation of the paper. We appreciate the recognition of the work’s relevance, the simplicity and generality of the proposed architectural modifications, and the thoroughness of the empirical comparisons. We are also glad that the introduction of the new plasma-physics benchmark was found valuable.
>
> **Weaknesses**
>
> [W1] Writing style
>
> We thank the reviewer for this helpful observation. In the revised manuscript, we refined Section 4 to make key ideas more explicit by stating important points upfront in each subsection. We also improved the logical flow of the methods and simplified the descriptions of notations and methodological details. We hope these changes improve readability and focus while keeping the technical content unchanged.
>
> [W2] Manuscript organization (Related work)
>
> We thank the reviewer for this helpful suggestion. In the revised manuscript, the Related Work section has been moved to appear immediately after the Technical Background section, before the Methods section. This reorganization provides clearer context for the proposed approach and improves the overall flow of the paper.
>
> [W3] Treatment of the second benchmark
>
> We thank the reviewer for the helpful feedback. We have expanded and restructured the Gray--Scott section in the main text to include the experimental results and discussion. We also have included a computational and memory efficiency study in the appendix, comparing the proposed and baseline models, and an additional experiment investigating varying spatial resolutions to demonstrate robustness across different discretizations. Together, we hope that these changes provide a more complete evaluation of the Gray--Scott system.
>
> [W4] Additional benchmark
>
> We are working to provide additional results before the author-reviewer discussion deadline.
>
> [W5] More qualitative assessment
> We thank the reviewer for this suggestion. To provide qualitative insight, we added an experiment varying the electron--neutral collision frequency \(\nu_m\) = (1e8, 2e8, 4e8) in the CCP system to examine the effect of coupling strength. All models show increasing nRMSE with higher \(\nu_m\), while the proposed hpFNOx remains the most stable and accurate (nRMSE: 0.015–0.018). This behavior aligns with the physical interpretation that stronger collisional coupling suppresses transport and creates stiffer, more localized structures that are harder to approximate spectrally. The results thus illustrate how coupling intensity affects learning difficulty and demonstrate that the proposed model remains robust across different coupling regimes.
>
> [W6] Typo and equation labeling
>
> Typo: Fixed. Thank you!
>
> Equation labeling: Regarding equation numbering, we originally omitted labels for brevity and numbered only the equations referenced in the text. In the revised version, we have added equation numbers throughout for easier cross-referencing.
>
> **Questions**
>
> [Q1] Sensitivity Hypernetwork
>
> We thank the reviewer for this valuable question. In the original implementation, we followed the design used in [Dupont et al. "From Data to Functa ...", ICML 2022], where the hypernetwork is implemented as a single linear layer that maps the physical parameter vector to layer-wise shift vectors.
>
> Per the reviewer’s request, we conducted additional experiments exploring alternative hypernetwork configurations with varying depth and width. Specifically, we tested 1- and 2-layer networks (width 256) and a 1-layer network (width 128), which produced nRMSE values in the range of 0.0171–0.0179, compared to 0.0154 for the linear baseline. Deeper or narrower configurations, such as a 3-layer network with width 256 (nRMSE: 0.0200) and width 32 (nRMSE: 0.0230), resulted in mild performance degradation.
>
> These results indicate that hpFNO performance is relatively insensitive to moderate architectural variations in the hypernetwork, while deeper or overly compact designs slightly affect stability. Overall, the single linear-layer configuration remains the most efficient and reliable choice.

---

> > ### Author Response · Authors · 2025-11-21
> > **Author rebuttal (2/2)**
> >
> > [Q2] Different modulation: $W_\ell(\mu)$
> >
> > We thank the reviewer for this question. While parameterizing $W_\ell(\mu)$ is a possible alternative, we find that modulating activations through additive shifts provides a more efficient and stable mechanism. Shift modulation introduces fewer additional parameters, preserves the shared spectral kernels of the base FNO, and maintains smoother optimization behavior.
> >
> > Empirically, we implement a variant that combines parameter-dependent weight modulation following the formulation suggested in the HyperFNO paper (suggested by the reviewer UVnJ, please refer to Eq. (17) in the revised PDF) along with the shift modulation.
> >
> > We test this design on the reaction rate benchmark and do not observe improvement with the weight-modulation variant (nRMSE: 0.0157) compared to the shift-modulation design (nRMSE: 0.0154). Given the similar performance and higher parameter cost, we retain the additive shift formulation for its simplicity and stability.
> >
> > [Q3] OOD test
> >
> > We thank the reviewer for this question. As discussed in paragraph “Study on the out-of-distribution samples,” in Appendix, we have examined this behavior to some extent and, in the revision, expand the analysis by adding visualizations showing model performance as $\mu$ moves away from the in-distribution (training) range.
> >
> > In general, we believe that extrapolation behavior is largely problem-dependent/parameter-dependent. In the tested reaction-coefficient scenario, all methods show a gradual increase in nRMSE as $\mu$ deviates from the training range, which is expected and consistent with typical data-driven surrogate models. The objective of our approach is to make this degradation less pronounced while keeping the model lightweight and easily deployable across different scenarios. Although larger-scale models (e.g., foundation models) could further improve extrapolation, our focus is on achieving a favorable trade-off between efficiency and generalization.
> >
> >
> > [Q4] Coupling in spatial domain
> >
> > We thank the reviewer for this question. Prior works on coupled or multiphysics neural operators (e.g., CFNO, CMWNO, CODA-NO) typically introduce coupling by employing multiple operator instances and exchanging information through hidden representations. These approaches effectively capture cross-variable dependencies but result in larger models and higher computational cost.
> > In contrast, our method performs coupling directly in the Fourier domain using a single operator instance, where spectral representations of the variables are mixed through a lightweight encoder–decoder module. This design retains the efficiency and scalability of the original FNO while enabling meaningful cross-variable interaction.
> > A more systematic way to incorporate explicit spatial mixing would be through localized operator formulations (e.g., [Liu-Schiaffini, et al, "neural operators with localized integral and differential kernels", ICML, 2024]), which provide spatially local kernels for capturing fine-grained interactions. However, such an extension would require substantial architectural modifications and is beyond the scope of this work.
> >
> > [Q5] Scalability w.r.t. the problem dimension and the number of variables
> >
> > We thank the reviewer for this question. Existing coupled neural operator methods such as CMWNO and CFNO generally increase the number of operator instances with the number of coupled variables. Each variable is handled by a separate operator stream, and information is exchanged between them through hidden representations. As far as we know, these approaches have not proposed mechanisms for efficiently mixing more than two variables.
> >
> > In contrast, our design keeps a single FNO instance and introduces coupling through a lightweight encoder–decoder operating in the Fourier domain. This keeps the kernel tensor size identical to that of a single-variable FNO, which is important for maintaining efficiency since the kernel scales as a 3D or 4D tensor in higher dimensions. The encoder–decoder adds modest parameter overhead but does not change the spectral kernel dimensionality.
> >
> > For 2D and 3D PDEs, the scaling behavior is similar to that of standard FNOs, where kernel dimensionality increases with spatial dimension. Both our method and FNO-family scale similarly in this regard.
> >
> > ------------
> > We thank the reviewer again for the positive evaluation and constructive feedback, as well as for the encouraging rating. The comments were very helpful in improving the manuscript. In the revision, we refined Section 4 for clearer exposition, improved the writing style to present key ideas more directly, expanded the Gray--Scott benchmark into the main text, added qualitative and scalability studies, and included new analyses on hypernetwork configurations and modulation strategies. We believe these revisions address all raised concerns and substantially enhance both the clarity and technical depth of the paper.

---

> > > ### Author Response · Authors · 2025-11-25
> > > **Additional results**
> > >
> > > [W4] Additional benchmark
> > >
> > > We again thank the reviewer for this valuable comment. We have now included a new 2D chemically reacting flow problem (Appendix F) to further demonstrate the generality of the proposed approach beyond what was originally covered. This problem involves four coupled physical variables: temperature, pressure, and two reacting species. These variables interact through nonlinear advection, diffusion, and reaction processes, introducing strong coupling and a more realistic and challenging test case.
> > >
> > > The new results, presented in Appendix F (Figure 18), show that the proposed models (FNOx and hpFNOx) consistently outperform baseline methods in both accuracy and computational efficiency. These findings indicate that the proposed extensions generalize effectively to higher-dimensional, multi-variable PDE systems, extending their demonstrated applicability.

---

> > > > ### Comment · Reviewer_6Lqg · 2025-11-25
> > > >
> > > > Thanks for the replies and clarification. For now, I think my score is appropriate. I'll keep monitoring progress on this paper.

---

> > > > > ### Author Response · Authors · 2025-11-26
> > > > > **Thank you!**
> > > > >
> > > > > We thank the reviewer again for the positive evaluation and for their continued engagement with the paper.

---

### Official Review · Reviewer_FkHd · 2025-11-01

**Soundness:** 4
**Presentation:** 2
**Contribution:** 3
**Rating:** 4
**Confidence:** 5

**Summary:**

This paper extends the Fourier Neural Operators (FNOs) to handle coupled and parameterized PDEs by introducing new architectures. The authors propose designs that model parametrized dynamics and coupled systems and evaluate it on a newly developed 1D capacitively coupled plasma and Gray-Scott dataset. Experimental results demonstrate that the proposed hpFNO models achieve higher predictive accuracy compared to existing FNO variants.

**Strengths:**

* The introduction clearly motivates the need for efficient operator learning on coupled and parametrized PDEs.
* The proposed pFNO, and hpFNO architectures aim to improve representations for coupled and parametrized PDEs, and the results on the benchmarks are promising.
* The ablation study thoughtfully analyzes the effects of design components, showing careful empirical work.

**Weaknesses:**

1. Lack of architectural visualization : The paper introduces multiple new and extended operators, but there is no schematic figure summarizing the overall architecture.
2. Unclear explanation for “best 5 of 10 runs” in Table 2 : Why is this criterion used for $T_{in} = 2$ and $T_{in} = 1$?
3. Sparse content in Section 4.3 : This section contains only a brief summary and refers the reader to the appendix. Including a compact summary table would be better for understanding cross-domain generalization.

**Questions:**

1. What exactly does “acc” in Figure 3 represent? It seems the lower is better, so 'acc' might be misleading.
2. How many CCP benchmark data were generated, and how were they split into training/test sets?

---

> ### Author Response · Authors · 2025-11-21
> **Author rebuttal**
>
> We thank the reviewer for the constructive feedback and for recognizing the strengths of our work. We appreciate the positive comments regarding the clear motivation, the design of pFNO and hpFNO architectures, the promising results on the benchmark datasets, and the careful empirical analysis in the ablation study.
>
> **Weaknesses**
>
> [W1] Additional visualizations to better illustrate the proposed architecture.
>
> We agree that visual clarification improves readability. In the revised version, we have added a set of diagrams (Appendix A.3) that illustrate both the parameterized and coupled extensions of FNOs and compare them visually with the original FNO architecture. These diagrams highlight where the modulation and coupling mechanisms are introduced relative to the baseline model.
>
> [W2] Unclear explanation for “best 5 of 10 runs” in Table 2: Why is this criterion used for T_in=2 and T_in=1?
>
> We appreciate the reviewer’s comment. The criterion “best 5 of 10 runs” was used because the training becomes unstable when the input window size is very short (T_in=2 or T_in=1). In these cases, some random initializations lead to non-convergent training runs. To obtain a reliable estimate of the model’s achievable performance, we repeated the experiments with 10 different random seeds and reported the results averaged over the five best performing runs. For all other configurations, which were stable, we used five random seeds as originally described.
>
> We have added a footnote in the main body to clearly deliver the intention, which reads:
>
> > For $T_\text{in}=\{2,1\}$, training instability was observed across some random seeds; due to this, we trained models with 10 random seeds and report the average over the five runs with the higher test prediction accuracy.
>
> [W3] Summary table in Section 4.3
>
> We thank the reviewer for this helpful comment. In the revised version, we expanded Section 4.3 to make it more self-contained  by adding a compact summary table (Table 4 in the revised version) that consolidates the design choices discussed in Sections 4.1–4.2 and Appendix A.1. The table summarizes how each architectural component (lift, Fourier layers, spectral coupling, and projection) can be configured for multi-variable and parameterized systems. Due to the current page limit, the table is placed in the Appendix, but we plan to move it into the main body once the additional page becomes available upon acceptance. We believe that these changes improve the clarity and completeness of Section 4.3.
>
> **Questions**
>
> [Q1] 'acc' in Figures
>
> We thank the reviwer for pointing this out. Our original intention is to indicate prediction accuracy. To avoid confusion, we have updated the figure labels to use “nRMSE” instead of “acc” throughout the revised manuscript.
>
> [Q2] How many CCP benchmark data?
>
> We thank the reviewer for this question. For the experiments shown in the main body of the manuscript, we generated three distinct datasets corresponding to the three 1D parameter cases (reaction rate, driving voltage, and ion mass). Each dataset contains 100 trajectories, each representing one parameter configuration. Following the common practice (e.g., in the original FNO paper), we used a 90:10 split for training and testing, respectively.
>
> We also revised the manuscript to make this clearer.
>
> -----------------
>
> We again thank the reviewer for their careful reading and detailed feedback. We hope that our revision addresses the identified concerns in the revised manuscript, including clarifying model visualizations, improving figure and table annotations, and expanding Section 4.3 with a summary table. We hope these revisions resolve the raised issues and make the paper clearer and more self-contained.

---

> > ### Comment · Reviewer_FkHd · 2025-11-21
> > **Update After Author Response**
> >
> > I would like to thank the authors for their rebuttal. The responses addressed all of my concerns clearly and satisfactorily.
> >
> >
> > I appreciate the additional clarifications and improvements, which significantly strengthened the paper. I have updated my rating accordingly.

---

> > > ### Author Response · Authors · 2025-11-21
> > > **Thank you!**
> > >
> > > We sincerely thank you for taking the time to re-evaluate our paper. We believe the revisions, especially the added visualizations and the summary, have significantly improved its clarity.

---

### Official Review · Reviewer_tkqC · 2025-11-03

**Soundness:** 3
**Presentation:** 3
**Contribution:** 3
**Rating:** 6
**Confidence:** 2

**Summary:**

1. The paper is well-written, easy to follow, and clearly motivated.

2. The experimental results are impressive, demonstrating a 55-72% reduction in error over strong baselines and confirming the value of the principled design.

3. I am not an expert in this area and will defer to the other reviewers for their expert opinions on the technical contribution

**Strengths:**

1. The paper is well-written, easy to follow, and clearly motivated.

2. The experimental results are impressive, demonstrating a 55-72% reduction in error over strong baselines and confirming the value of the principled design.

**Weaknesses:**

The manuscript would benefit from additional visualizations, such as a diagram illustrating the model architecture, to help readers better understand the proposed method.

**Questions:**

What is the key difference between your method and the baselines, and why does your approach work?

---

> ### Author Response · Authors · 2025-11-21
> **Author rebuttal**
>
> We thank the reviewer for their positive assessment of the paper’s motivation, clarity, and presentation, and for recognizing the performance improvements demonstrated in the experiments.
>
> **Weakness**:
> Additional visualizations to better illustrate the proposed architecture.
>
> We agree that visual clarification improves readability. In the revised version, we have added a set of diagrams (Appendix A.3) that illustrate both the parameterized and coupled extensions of FNOs and compare them visually with the original FNO architecture. These diagrams highlight where the modulation and coupling mechanisms are introduced relative to the baseline model.
>
>
>
> **Questions**:
> The key difference between your method and the baselines; why does your approach work?
>
> We thank the reviewer for this question. Our work differs from prior neural operator approaches by introducing minimal, component-wise architectural modifications that extend the FNO to model both parameterized and coupled PDE systems efficiently.
>
> For parameterized dynamics, our approach employs a shift-modulation mechanism, where a compact hypernetwork generates parameter-dependent shifts that modulate internal activations rather than regenerating the entire layer weights. Explicitly conditioning neural operators on known physical parameters remains relatively underexplored in the literature. Compared to prior work such as HyperFNO [Alesiani et al., "HyperFNO: Improving the Generalization Behavior of
> Fourier Neural Operators", MLPS workshop, NeurIPS, 2022], which uses a large hypernetwork to infer most model parameters, our design achieves comparable flexibility with substantially fewer parameters and greater training stability, leading to improved accuracy across all tested benchmarks.
>
> For coupled systems, our method introduces spectral-domain coupling through a single FNO instance, where spectral representations of multiple fields interact via a lightweight encoder–decoder structure. In contrast, existing coupled operator frameworks such as CMWNO or CFNO rely on multiple operator streams and exchange hidden representations between them, increasing model size and computational cost. Our approach achieves similar or better accuracy while keeping the kernel tensor and parameter count nearly unchanged from the base FNO.
>
> Overall, these targeted modifications make the proposed models both compact and expressive, leading to consistent improvements over stronger baselines such as HyperFNO, CFNO, and CMWNO without sacrificing efficiency.
>
> -----------------
>
> We again thank the reviewer for their careful reading, detailed feedback, and positive evaluation. We hope that our revision addresses the identified concerns in the revised manuscript.

---

### Author Response · Authors · 2025-11-21
**Global response**

## Global response
We thank all reviewers for their thoughtful and constructive feedback. The comments helped refine both the presentation and the experimental analysis and we believe the manuscript has been improved significantly thanks to the comments. We kindly invite the reviewers to consult the revised PDF for updated figures, tables, and additional experiments. While a few extended studies are still underway, we believe the current version thoroughly addresses the key issues raised in the reviews.

**Strengths**: We appreciate that the reviewers recognized the paper’s clear motivation, a minimal yet principled way of extensions to FNOs, strong empirical performance, systematic ablation, and introduction of a new plasma-physics benchmark. Several reviewers also noted that the modular, lightweight nature of our proposed design is a strength that makes the method practical.

**Revision we made**:

*Editorial and structural revision*

- **Section organization**: Moved the Related Work section to follow the Technical Background and improved transitions throughout the Methods section to maintain a more focused narrative.
- **Conciseness and clarity**: Revised several paragraphs (particularly in the Introduction and Section 4) to avoid meandering writing and to emphasize the core ideas more directly.
- **Visual aids**: Added new architectural diagrams in Appendix A.3 comparing the proposed parameterized and coupled FNOs against the original FNO for easier interpretation (Due to page limits, the figures are currently in the Appendix but can be moved to the main body  upon acceptance.)
- **Notation and readability**: Corrected typographical errors, added equation numbering, and clarified notation used across sections.
- **Summary table**: Expanded Section 4.3 with a compact table summarizing architectural configurations (Due to page limits, the table is currently in the Appendix but will be moved to the main body upon acceptance.)
- **Restructuring Gray--Scott subsection**: Section 5.3 now includes the detailed performance table, analysis, and discussion that were previously in the Appendix.
- **Clarified training protocol**: Explained the “best 5 of 10 runs” criterion for short input windows \(T_in\)={2,1} and updated Table 2’s caption accordingly.

*Experimental and analytical revisions*

- **HyperFNO experiments**: Added new comparisons with HyperFNO [Alesiani et al., "HyperFNO: Improving the Generalization Behavior of
Fourier Neural Operators", MLPS workshop, NeurIPS, 2022]; hp-variants still demonstrate improved performance over a new baseline.
- **Hypernetwork study**: Conducted additional experiments on hypernetwork configurations with varying depth and width, showing that hpFNO performance is relatively insensitive to moderate changes while the linear design remains most effective.
- **Alternative modulation**: Implemented and evaluated a weight-modulation variant, found no improvement over the proposed shift-modulation approach.
- **OOD test description**: Added further discussion on out-of-distribution (OOD) behavior and problem-dependent extrapolation trends.
-  **Coupling effect**: Added an experiment varying the electron–neutral collision frequency in the CCP system to analyze model robustness under different coupling strengths.
-  **Additional Gray--Scott experiments**: added an efficiency comparison (computational and memory cost) in the Appendix and an additional experiment on varying spatial resolutions showing consistent accuracy across discretizations.

We thank the reviewers again for their valuable and constructive feedback, which has directly strengthened both the presentation and the technical depth of this work. Below, we provide detailed responses to individual comments.

---

### Author Response · Authors · 2025-11-30
**Executive summary of the current discussion status**

Dear Area Chair,

We would like to provide brief context regarding the review timeline. Importantly, all reviewer discussions, clarifications, and the rating updates that informed the current decision were completed no later than Nov 26th, fully prior to the OpenReview incident (Nov 27th). We are grateful to the reviewers for their timely engagement, active discussion, and constructive feedback throughout the process.

We would now like to summarize the revisions made in response to the reviews and how the reviewers reacted to them. After the rebuttal and major updates, two reviewers updated their ratings from 4 to 6, resulting in **6 / 6 / 6 / 6**.

Below we highlight the key changes in the order they appear in the revised manuscript.

1. **Significant restructuring and rewriting for clarity**
Multiple reviewers noted that earlier drafts contained meandering writing and less clear transitions. In response, we performed a major editorial overhaul, including reorganizing the Introduction, moving the Related Work section earlier, tightening Section 4 and Section 5 for better logical flow, and clarifying notation throughout. Reviewers acknowledged the improved clarity and focus.

2. **Architectural clarity and visual aids**
Reviewers requested additional diagrams and clearer structural explanations of the proposed extensions. We added new architectural figures for the parameterized and coupled FNO designs, clarified operator blocks, and included a compact configuration summary table. Reviewers acknowledged that these additions substantially improved interpretability.

3. **Gray--Scott benchmark section restructuring**
The Gray--Scott section previously felt secondary. We moved the full performance tables, analysis, and discussion into the main text and added discussion on computations/resolution-variation experiments in Appendix. This resolved the reviewers’ concerns about completeness and presentation quality.

4. **Expanded experimental analysis**
Reviewers requested deeper ablations and broader empirical investigation. We conducted the following additional studies:

- **Hypernetwork depth and width sensitivity**: We tested multiple hypernetwork sizes and found hpFNO is robust, with the linear configuration performing best.

- **Stronger baseline, HyperFNO**: A reviewer asked how our method compares to HyperFNO [Alesiani et al., MLPS, NeurIPS 2022].   The updated results show that hpFNO continues to achieve lower prediction error while maintaining significantly lower computational/memory cost.

- **Weight-modulation alternative**: We implemented the HyperFNO-style weight modulation variant. It occasionally helps but does not consistently outperform shift modulation and requires careful tuning.

- **Multiplicative (scale) gating modulation alternative**: We evaluated amplitude-based gating. Naive gating was unstable; however, our stabilized formulation using $1 + \eta \text{tanh}(\gamma(\mu))$ produced reliable and sometimes improved results. In conclusion, we recommend shift modulation for consistent performance, though scale modulation can help when tuned carefully. (*This is indeed a new addition finalized after the last reviewer discussion on Nov 26th, included here for completeness.*)

- **Out-of-distribution parameter behavior**: We expanded OOD analysis showing smooth degradation and problem-dependent extrapolation trends.

- **Coupling-strength analysis in the CCP system**: We added additional experiments by varying electron–neutral collision frequency to assess robustness across coupling regimes. Fourier-domain coupling behaved consistently well.

- **Additional Gray--Scott analyses**: We added efficiency comparisons (runtime and memory) and spatial-resolution studies, demonstrating consistent accuracy across discretizations.

- **New benchmark: 2D chemically reacting flow with four coupled variables**:
Several reviewers asked about scalability to higher-dimensional or more complex PDE systems. We added a new 2D chemically reacting flow problem with four interacting fields. This benchmark clearly demonstrates that our coupled-FNO design scales effectively to higher-dimensional, multi-field systems.

Reviewers responded positively to these additional experiments, and one reviewer raised their rating to 6 as a result.

----------------

Overall, with these updates, two reviewers updated their scores, resulting in unanimous 6. We hope this summary supports your evaluation and facilitates the decision process. We sincerely appreciate your time and consideration.

---

### Meta-Review · Area_Chair_Sc5L · 2026-01-13

**Summary:**

The reviewer generally indicate that:
1. The work is well motivated and addresses important engineering problems -- how to deal with coupled equations and finite-dimensional parameters within them.
2. Numerical experiments show consistent improvement among all benchmarks and the chosen benchmarks form a good representation for the current state-of-the-art.
3. The method is simple to understand and implement and yet is very effective.
4. Some of the writing and presentation of the results is not clear and some figures and tables need to be re-worked.

**Reviewer Concerns:**

The authors have re-written and re-organized parts of the paper, included new figures and baselines, more carefully analyzed results from existing experiments, and added various ablations to motivate their choices. All reviewer concerns have been addressed.

**Reviewer Scores:**

All reviewers have raised their scores to a 6.

---

### Decision · Program_Chairs · 2026-01-26

Accept (Poster)